# Towards Prototype Conformity Loss Functions for Better Out of Distribution Detection in Traffic Sign Image Classification

## Abstract

Deep neural networks (DNNs) generate overconfident outputs even in case of miss-detections caused by abnormal data. Consequently, this can lead to unreliable classifications and, thus, potentially lead to issues in safety-critical applications such as automated driving systems. Recent works propose to detect such anomalous data based on probabilistic methods derived from the DNN's internal activation functions, such as the convolutional neural networks (CNN) backbones. This paper shows that such CNNs cannot semantically disentangle similar classes when trained with conventional cross-entropy loss functions, leading to poor out-of-distribution (OOD) detection while applying probabilistic methods for such a purpose. Therefore, we propose to apply the prototype conformity loss (PCL) function from the literature and show that such a contrastive learning method leads to better OOD detection for traffic sign classification. Furthermore, we propose two novel variations of the PCL, namely weighted PCL (WPCL) and multi-scale PCL (MSPCL), which group similar classes and force the DNN to disentangle them from each other. In contrast to existing contrastive OOD detection literature, we do not rely on complex input transformations or augmentations. We perform our experiments on multiple DNNs and two traffic sign classification datasets, which we test against multiple OOD data sources, such as adversarial and non-adversarial augmentation and real-world OOD data. Based on that, we demonstrate that our PCL variations can achieve superior results in OOD detection when the training dataset includes various similar classes.

## 1 Introduction

The safety of deep neural networks (DNNs) has been a significant topic within automated driving research in recent years Aravantinos & Schlicht (2020); Sämann et al. (2020); Schwalbe et al. (2020); Schwalbe & Schels (2020); Willers et al. (2020). DNNs often appear as black-box modules and therefore pose a challenge to the safety evaluation of the overall system that relies on their function. To ensure the safe functionality of DNNs, one can study corner cases in which the DNNs fail to predict correctly. However, identifying such corner cases for DNN-based modules, especially at runtime, is not easily feasible due to their aforementioned complexity and resulting decision uncertainty.

Furthermore, DNNs are sensitive to unseen data. Concisely; this means that any small change in the input data can cause a covariance shift from the data the DNN is trained with and can potentially lead to unreliable predictions by the DNN, which can be extremely difficult to track in practice Yang et al. (2021). Such a covariance shift in inputs could stem from different sources such as adversarial data generated by adversarial attack methods Goodfellow et al. (2014; 2015); Fawzi et al. (2016); Eykholt et al. (2017); Samangouei et al. (2018); Sitawarin et al. (2018); Brown et al. (2018); Soll (2019); Feng et al. (2021), which aim at deviating the DNN decision in certain directions. Moreover, other unseen changes in the input data include different natural causes or sensor failures, e.g., destruction and color changes of the objects in the real world due to vandalism or aging. In the case of traffic sign classification, different adversarial attacks are introduced that are deployable to the real world in the form of adversarial stickers to be put on the traffic signs. Despite

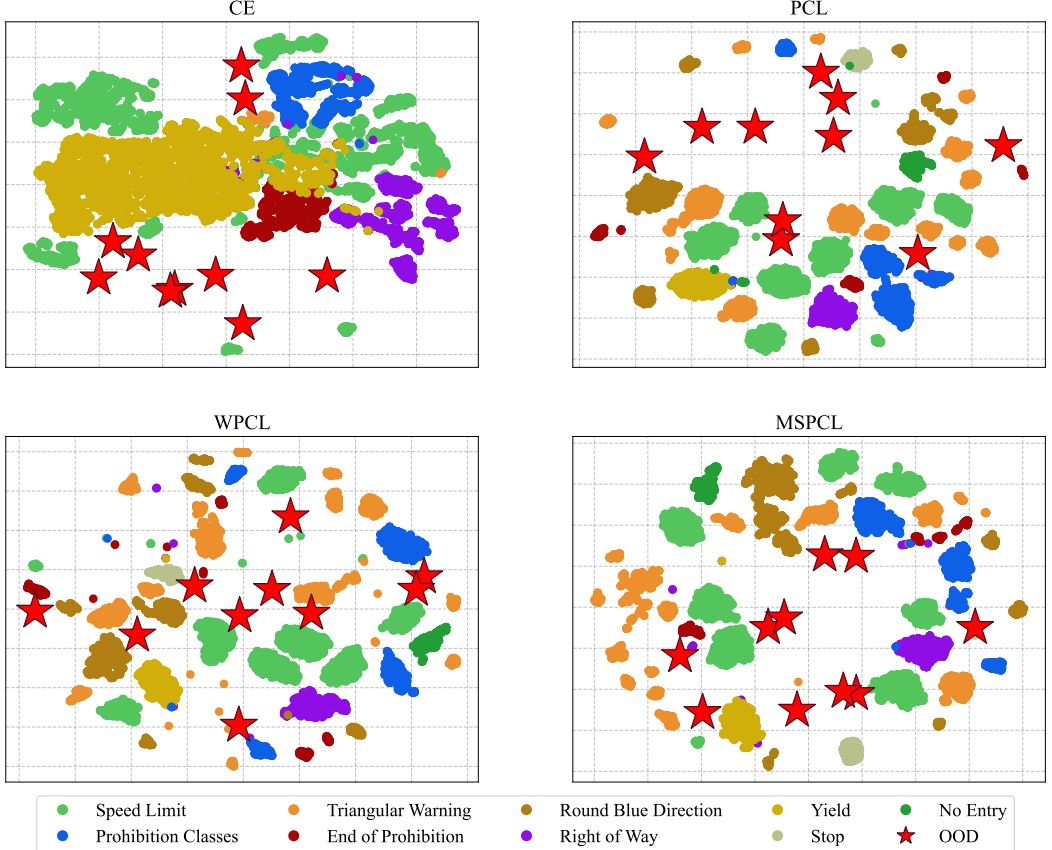

Figure 1: The t-SNE van der Maaten & Hinton (2008) visualization of different optimization methods based on the filter activations extracted from the WideResNet50 Zagoruyko & Komodakis (2016) on the GTSRB dataset Stallkamp et al. (2011). The optimization methods include cross-entropy (CE), original prototype conformity loss (PCL) Mustafa et al. (2021) and our proposed two new losses i.e, weighted prototype conformity loss (WPCL), and multi-scale prototype conformity loss (MSPCL). The colors represent the grouping that has been done to the GTSRB dataset presented in sub-section 4.1. Accordingly, multiple instances of a color represent all the classes belonging to the same group. The plot indicates that our proposed methods achieved a more sparse projection of the activations, leading to improved OOD detection. In the plot, we refer to "Clean" samples as the original data of each class, and "OOD" refers to the attacked data of certain classes.

their performance, DNNs often generate false predictions with high confidence, which may lead to hazardous events if not detected and compensated by other mechanisms of the safety critical systems. Therefore, it is essential for the system to actively recognize when to rely on the decisions made by the DNN module and when not to, thereby increasing the overall robustness of the system.

Accordingly, we refer to out-of-distribution (OOD) based on the definition provided by Yang *et al.* Yang et al. (2021) as the covariance shift occurred to the input data ($x$) while having the same semantic meaning ($y$), and any other distributional shift in the input data causing a semantic shift as well, such as unforeseen traffic sign classes. Therefore, our catalog of OOD data includes real-world traffic sign samples with occlusions due to different factors such as dirt, snow, etc., adversarial augmentations leading to the miss-classification of the input data, and non-adversarial augmentations resembling vandalized traffic signs. Based on that, our motivation is to optimize the OOD detection methods based on latent space activations generated from the clean training data sets as their primary operational design domain (ODD) and consider any shift in the input space as OOD. To better achieve this, we aim to encourage the DNNs to provide better latent activations to help the OOD detectors perform better.

In this paper, we utilize a prototype conformity loss (PCL) alongside the conventional cross-entropy loss to train traffic sign classifiers for better out-of-distribution (OOD) detection. We show that when combined with the standard OOD detection methods from the literature, training the classifiers with PCL would lead to better OOD detection while maintaining, if not outperforming, the baseline accuracy. Furthermore, we show that in particular classification tasks, such as traffic sign image classification, where there are various classes with strong visual similarity, the PCL can also be improved to tackle such a similarity and lead to even better OOD detection. Such visual similarities can be found in the speed limit sign classes, i.e., 30, 50, 70, and 80 km/h, wherein the overall shape of the traffic signs is identical except for one or two digits that distinguish one sign from the others.

An illustration of our PCL training approach can be found in Figure 2. As shown in the figure, during the DNN training, the feature representations are extracted from one or more layers of the DNN, wherein different PCL losses are calculated. These losses are then aggregated and combined with the main cross-entropy (CE) loss of the classifier to force it to better disentanglement the embedded space representations. A hypothetical illustration of the possible features can be found in Figure 3. It shows that the DNNs trained with conventional CE loss would generate feature space representations for various classes that fall very close to each other. We show that this can be improved by utilizing the PCL while training the DNNs for the common classification tasks.

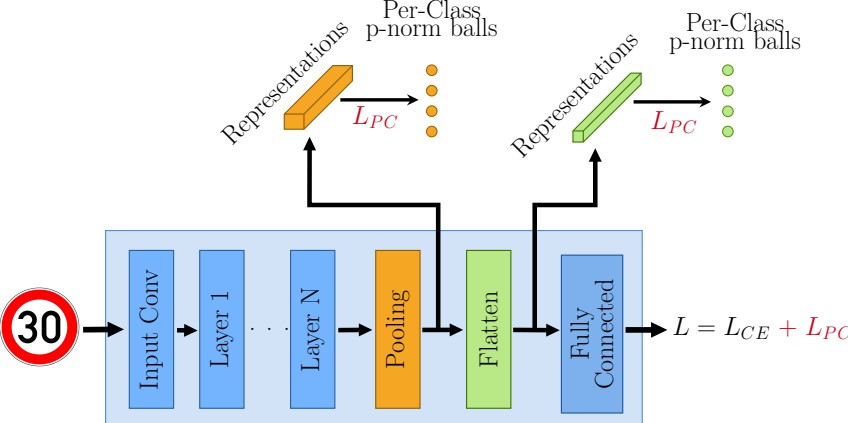

Figure 2: The overview of a common DNN utilized with PCL loss along with the CE loss. During the training, we first compute the conformity losses at two different locations (before and after flattening the embedded representations) and then average them together in the final CE loss. Our WPCL and MSPCL employ the same representations for their computation but differ compared to the original PCL loss, further improving the disentanglement for similar classes.

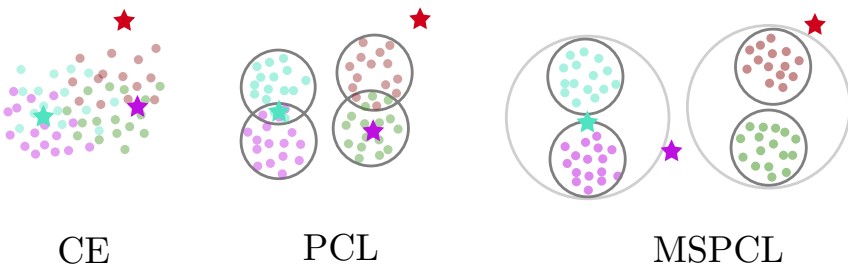

CE              PCL              MSPCL

Figure 3: A simplified illustration of filter activations in different training setups including CE: the conventional cross-entropy loss; PCL: the original PCL Mustafa et al. (2021) which can disentangle dissimilar classes, while similar classes would still lead to having overlapping p-norm balls; MSPCL: our MSPCL as a variation of PCL, where a p-norm ball is defined for each group of similar classes (light gray), and individual classes p-norm balls (dark gray). Different colors depict different classes, dots depict clean data samples, and stars depict adversarial or other outliers, leading to miss-classifications.

However, as traffic sign classification involves distinguishing classes with strong visual similarity, we propose two novel variations of the original PCL, namely weighted PCL (WPCL) and multi-scale PCL (MSPCL) to force the DNNs to generate sparse feature space representations for such similar classes. In our experiments, we show that such an approach outperforms the CE and PCL across multiple DNNs and OOD detection methods. An actual illustration of our hypothesis can be found in Figure 1. This figure illustrates the activations extracted from the WideResNet50 Zagoruyko & Komodakis (2016) classifier while trained with all four training methods. The activations include clean test samples with colorful dots and attacked images as OOD with red star markers. Each color in this figure represents either a group of similar classes or a single class if it does not belong to any group. The groups are discussed in detail in Section 4.1. It shows that the DNN generated activations for all the clean samples for the common CE training, which fall very close to each other. Furthermore, in contrast to the aforementioned problem, our proposed PCL variations lead to better feature disentanglements compared among similar classes to CE and the original PCL training method. It can be observed that the clusters of similar classes that share the same colors fall very close to each other for both the CE and PCL training methods. This is mitigated in our training approaches by pushing such clusters away from each other, which is the purpose of such a grouping.

Accordingly, our contributions are as follows:

- Propose to apply the contrastive PCL loss function from literature to enhance feature-based OOD detection algorithms.

- Propose weighted prototype conformity loss WPCL and multi-scale prototype conformity loss MSPCL to tackle the problem of very similar classes disentanglement in the embedded space.

- Conducted experiments over six DNNs, two traffic sign classification datasets, four image augmentation techniques, and six OOD detection methods. Our MSPCL outperforms the CE and PCL by 3.36% and 3.20% in AUROC and 5.15% and 1.00% in FPR across six DNNs and five ODD methods in our challenging GTSRB benchmark.

The rest of this paper is organized as follows: we review the related work in Section 2; our proposed methods are introduced in Section 3; the experiment setup is explained in detail in Section 4; the results are discussed in Section 5; and the paper is concluded with the key achievements and possible future work in Section 6.

## 2    Related Work

During recent years, ensuring the safety of DNN within safety-critical systems has received great attention from the research community Aravantinos & Schlicht (2020); Sämann et al. (2020); Schwalbe et al. (2020); Schwalbe & Schels (2020); Willers et al. (2020); Heyn et al. (2023); Goodloe (2022). This is due to

the fact that the output of DNNs is sensitive to the changes in the input data including adversarial augmentations Goodfellow et al. (2014); Kurakin et al. (2017); Eykholt et al. (2017); Moosavi-Dezfooli et al. (2017); Samangouei et al. (2018); Brown et al. (2018); Soll (2019); Feng et al. (2021); Hu et al. (2022) or other factors including damage, vandalism, etc. Bielik et al. (2020); Magnussen et al. (2020); Wali et al. (2019). This poses a significant safety concern to DNNs deployment to the real world applications.

The aforementioned adversarial attacks can be divided into two categories. In the first category, attacks aim to introduce malicious content to all images (with or without any specific target class). Such methods require access to the images before DNN inference to augment them with the adversarial content Madry et al. (2018); Soll (2019). On the other hand, the second category includes patch-based attacks to the detectable objects (e.g., traffic signs in our case), where the attacks are optimized over only a small patch of the image by adding adversarial stickers to partially occlude such objects Eykholt et al. (2017); Brown et al. (2018); Feng et al. (2021); Sitawarin et al. (2018).

Eykholt *et al.* Eykholt et al. (2017) introduced Robust Physical Perturbations (RP2), which is a one-to-one attack (one source to one target class) for targeting traffic sign classifier i.e., DNN. RP2 is optimized to mislead the classifier from the source class to a target class. The generated patches were applied to traffic signs of the real world. Similarly, Brown *et al.* Brown et al. (2018) introduced the Adversarial Patch (AP), which is a many-to-one attack (multiple source classes to a single target class). The goal of this attack is to deviate the attacked DNN's detection from all source classes to a single target class. Unlike RP2, AP is location, rotation, and scale-invariant, which allows it to be applied anywhere on the object with different rotations and scales. Similar to Eykholt et al. (2017), Sitawarin *et al.* Sitawarin et al. (2018) proposed an OOD adversarial attack for traffic sign dataset. Recently, Bayzidi *et al.* Bayzidi et al. (2022) discussed the main challenges of applying realistic stickers to traffic signs with augmentation techniques. They proposed a multi-step approach of conventional image processing techniques to adjust the sticker overlays to the individual traffic sign images with varying properties.

While adversarial attacks and other augmentation methods have proven to be able to mislead the state-of-the-art DNNs into highly confident wrong predictions, detecting examples deviating from the training distribution as outliers have been extensively studied in the literature as well Feinman et al. (2017); Lee et al. (2018); Ma et al. (2018); Pang et al. (2018); Xu et al. (2018); Papernot & Mcdaniel (2018); Carrara et al. (2019); Cohen et al. (2020). We can divide these approaches into two different branches. The first branch consists of the post-hoc approaches, which mainly rely on an outlier scoring function based on the output of a trained DNN. Such approaches include confidence scores, uncertainty-based approaches, or functions like the energy applied on the DNN outputs. A simple baseline (MSP) Hendrycks & Gimpel (2017) by Hendrycks *et al.* detects outliers based on the maximum SoftMax value. The ODIN Liang et al. (2018) method by Liang *et al.* extends MSP by perturbing the DNN inputs. As an alternative to the maximum SoftMax criterion Hendrycks *et al.* Hendrycks et al. (2022) evaluated the maximum logit value (MaxLogit) and the KL divergence computed on the mean class-conditional logits. Lee *et al.* Lee et al. (2018) suggested the Mahalanobis distance as a detection metric that one can use to discover the outlier data points in latent space. Liu *et al.* Liu et al. (2020) propose the energy function as a metric based on the logits. The ReAct approach Sun et al. (2021) by Sun *et al.* extends the energy method by rectifying activations prior to logit computation. In ViM Wang et al. (2022) by Wang *et al.*, a statistical subspace projection is applied on the last layer weights to enhance energy-based detection. Recently, Sun *et al.* Sun et al. (2022) investigated the distance function of k nearest neighbors to score outliers.

The second branch consists of training-based detection introducing additional regularization objectives. These aim to either optimize a specific detection function or enhance post-hoc algorithms. This can be implemented by adding outlier data during training and enforcing separable score distributions for such samples and the training distribution Hendrycks et al. (2019), Liu et al. (2020), Tao et al. (2023). Moreover, others fit the features of a DNN to probabilistic models during training to better distinguish inlier and outlier feature embeddings at test time Du et al. (2022b), Du et al. (2022a). Following the recent advances in contrastive learning, Tack *et al.* Tack et al. (2020) proposed a training framework based on SimCLR Chen et al. (2020) using transformed images and a specific scoring criterion. Sehwag *et al.* Sehwag et al. (2021) similarly detect outliers by employing contrastive learning on transformed images and the Mahalanobis distance as a score. Ming *et al.* Ming et al. (2023) combine probabilistic modeling of features and contrastive

losses to create separable embeddings in the hyperspherical space. This leads to distant and compact latent embeddings for each class. One contrastive training objective that eliminates the need for identifying and applying suitable input transformations is the conformity loss Kapoor et al. (2020; 2021); Mustafa et al. (2021). Mustafa *et al.* Mustafa et al. (2021) proposed the prototype conformity loss (PCL) combined with the conventional cross-entropy loss function for training to increase robustness against adversarial and non-adversarial perturbations of DNNs. This is achieved by forcing the latent activations of different classes to be maximally separated from each other, a desirable property for OOD detection. Nevertheless, to the best of our knowledge, there are no OOD detection approaches that benefit from such an optimization based on only clean input samples without the need for additional transformations. Therefore, we extend the PCL to enhance OOD detection algorithms and propose two novel variations specifically designed for the OOD detection task.

## 3 Method

DNNs often produce confident output scores, even when misclassifying data. However, it is possible to detect if OOD data is inferred into the DNN under test. This is done by evaluating the distance of the bespoken data sample with the group of clean data samples from the training dataset using the latent activations of the DNN. Accordingly, multiple feature-based OOD detection methods have been proposed in recent literature. We want to emphasize that the DNN is trained only with clean data, and no augmented data, such as adversarial or noisy data, is involved during the training.

Furthermore, in most real-world applications, there can be overlaps or close activations of the training samples of different classes due to their semantic similarity. For example, in the case of traffic sign classification, the activations of speed limit classes would be close to each other, as their only distinguishing features are mostly one or two digits, such as 30km/h and 80km/h. This can lead to sensitivity of the DNNs against OOD and adversarial data on one side and poor OOD detection in latent space on the other side, as anomalous data samples might be projected very close to clean ones in latent space. Therefore, we argue that there is a need to alter the training process of such DNNs to also clearly separate the activations of different classes from each other. Consequently, different classes should have prototype p-norm balls with minimum or no overlaps.

To achieve this, we propose two novel loss functions based on the PCL Mustafa et al. (2021) to force the activations of different classes to be separated, which would support better OOD detection based on such activations. These include weighted prototype conformity loss (WPCL) and multi-scale prototype conformity loss (MSPCL). Firstly, we discuss the details of the conventional PCL in this section. Following that, we discuss the problems of applying PCL to real-world applications such as traffic sign classification, which involves various similar classes. Finally, we introduce our novel prototype conformity losses, which can outperform the standard PCL in OOD detection capability. For our ablation study, we apply multiple OOD detection methods to latent activations for different training variations, including normal cross-entropy, PCL, WPCL, and MSPCL training.

To make things easier to understand in the subsequent sections, let's first discuss some mathematical notations, which this section will consistently follow.

**Notation:** Let $\boldsymbol{x} = (\boldsymbol{x_i}) \in \mathbb{R}^{C \times H \times W}$ and $\boldsymbol{y} = \{1, ..., c\}$ denote an input-label pair, where $c$ denotes the number of classes in the dataset. We denote the DNN which consists of $L$ layers by $\mathbf{F}_\theta : \boldsymbol{x} \to \mathbb{O}$, where $\theta$ are the trainable parameters and $\mathbb{O}$ represents the classification output of the DNN. Further, the DNN outputs a feature representation i.e., $\boldsymbol{f} = \mathbb{O}_{L-1} \in \mathbb{R}^d$, which are used by the classification layer to perform multi-class classification and produce $\mathbb{O}$. The parameters of $\mathbf{F}$ can be represented as $\boldsymbol{W} = [\boldsymbol{w_1}, ..., \boldsymbol{w_c}] \in \mathbb{R}^{d \times c}$.

**Prototype Conformity Loss (PCL) Mustafa et al. (2021)**: As mentioned earlier, the goal of such a loss is to disentangle different classes in latent space. Specifically, it considers a p-norm ball $P$ for each class, wherein the goal of PCL would be to 1) minimize the possible overlaps among the hyper-balls of different classes, and 2) encourage all the data samples activations to fall into their respective class hyper-balls. This way, the centroids of the p-norm balls are learned automatically to maintain a maximum distance from each other, eventually leading to no overlap among them. Thus, for calculating the PCL loss, the proximity $p$ is

first calculated as follows:

$$p(\boldsymbol{x}_i, \boldsymbol{y}_i) = ||\boldsymbol{f}_i - \boldsymbol{w}^c_{y_i}||^2_2, \tag{1}$$

where $\boldsymbol{w}^c \in \mathbb{R}^d$ is the p-norm ball optimizable centroid of class $c$. Furthermore, the contrastive proximity i.e., *cp* between two classes can be calculated as follows:

$$cp(i,j) = ||\boldsymbol{f}_i - \boldsymbol{w}^c_j||^2_2 + ||\boldsymbol{w}^c_{y_i} - \boldsymbol{w}^c_j||^2_2. \tag{2}$$

Finally, the PCL is calculated as follows:

$$L_{PC}(\boldsymbol{x}, \boldsymbol{y}) = \sum_i \Big\{ p - \frac{1}{c-1} \sum_{j \neq y_i} \big( cp(i,j) \big) \Big\}, \tag{3}$$

As shown in the Equations 2 and 3, the second part for calculating the contrastive proximity of class $i$ with all the other classes is averaged over all the classes in the dataset. In the case of real-world applications such as traffic sign classification, this would lead to the minimum effect of such loss functions for classes with strong visual similarity, such as speed limit signs. For example, if $x_i$ belongs to the class 30km/h, it could lead to a high *cp* with similar classes, such as 50km/h or 80km/h. However, as the *cp* is averaged over all the classes, including dis-similar ones, it would be decreased and lead to projecting such similar classes close to each other in the latent space, despite being trained with such a proximity loss. Therefore, it is essential to distinguish the proximity loss calculated for similar classes from dis-similar ones so that the DNN is penalized more when projecting such similar classes close to each other. Based on this proposition, we propose two novel alternations of PCL to overcome this problem in the following.

**Weighted Prototype Conformity Loss (WPCL)**: As mentioned earlier, the visual similarity of certain classes would minimize the effect of the contrastive proximity *cp* depicted in the Equation 2, when they are averaged over all classes. Therefore, we propose to alter the *cp* with a hyper-parameter to force the similar classes with a stronger *cp*. Based on that:

$$WL_{PC}(\boldsymbol{x}, \boldsymbol{y}) = \sum_i \Big\{ p - \frac{1}{c-1} \sum_{j \neq y_i} \begin{cases} \lambda * cp & j \in \mathbb{S}^i \\ \\ cp & j \notin \mathbb{S}^i \end{cases} \Big\}, \tag{4}$$

where $\lambda > 1$ and $\mathbb{S}^i$ are the sets of all the similar classes that the current image $\boldsymbol{x}$ falls into, which can be defined manually for each dataset. This is specifically simple for traffic signs as they can be grouped based on their structural and visual similarities, such as shape, color, texture, etc. This way, if the activation $\boldsymbol{f}_i$ of image $\boldsymbol{x}_i$ falls into the p-norm balls of a similar class, it will be additionally penalized for the respective similar class compared to the other, non-similar, classes.

**Multi-Scale Prototype Conformity Loss (MSPCL)**: Similar to WPCL, we propose the MSPCL as an alternative to the original PCL to tackle the problem of overlapping hyper-balls of similar classes. In this method, we also enforce class similarity based on groups defined similarly to WPCL while considering that there might be single classes that do not belong to any group. For example, in the case of traffic sign classification, there can be groups of speed limit signs, triangular danger signs, and single-class groups, e.g. the stop sign. Derived from that, the PCL will be calculated in two scales: group scale and single class scale. Accordingly, if a class belongs to a group, such as the speed limit, or is a single class, such as a stop sign, then a proximity $p^g$ and a contrastive proximity $cp^g$ will be calculated on the group level as follows:

$$p^g(\boldsymbol{x}, \boldsymbol{y}) = ||\boldsymbol{f}_i - \boldsymbol{w}^g_{y_i}||^2_2 \tag{5}$$

and

$$cp^g(i,j) = ||\boldsymbol{f}_i - \boldsymbol{w}^g_j||^2_2 + ||\boldsymbol{w}^g_{y_i} - \boldsymbol{w}^g_j||^2_2, \tag{6}$$

where $\boldsymbol{w}^g$ is the p-norm ball for the group that the image class $\boldsymbol{x}_i$ belongs to. If the class does not belong to any group, it will only have a p-norm ball in the group scale. Otherwise, another PCL is calculated within each group with more than one class. This is determined based on Equations 1, 2 and 3 with the difference that the centers of the hyper-balls in this scale are subtracted from their respective group center so that

they are forced to adopt themselves to their respective group centroids. Finally, the $L_{MSPC}$ is calculated as follows:

$$MSL_{PC}(\boldsymbol{x}, \boldsymbol{y}) =$$
$$\sum_i \left\{ p^c + p^g - \frac{1}{c-1} \sum_{j \neq y_i} \left( cp^c(i, j) + cp^g(i, j) \right) \right\}, \quad (7)$$

where $p^c$ and $cp^c$ are calculated for each class, and $p^g$ and $cp^g$ are calculated for each group of classes. If a class does not belong to any group, only $p^g$ and $cp^g$ are calculated; therefore, $p^c$ and $cp^c$ will be zero. A simplified example of different training methods and their effect on encoding is illustrated in Figure 3. As shown in sub-figure (a) of this figure, the cross entropy training would lead to many overlapping clusters, making finding anomalous data samples challenging. The reason is that the probabilistic methods rely mostly on data distribution to compare novel data samples with and define them as inliers or OOD. In sub-figure (b), one can see an example of training with PCL, which leads to the better disentanglement of activation clusters of dissimilar classes but would lead to hyper-balls of classes with similar shapes still overlapping. In sub-figure (c), our MSPCL aims to disentangle in different scales and achieve better hyper-ball disentanglement and, thus, better OOD detection capabilities.

**Combined WPCL and MSPCL**: We also performed experiments on combining both of our proposed approaches to provide a better understanding of the joint effect on the training and the OOD detection results. To do so, we considered a similar weighting of the combined loss as follows:

$$COM_{PC}(\boldsymbol{x}, \boldsymbol{y}) = 0.5 * MSL_{PC}(\boldsymbol{x}, \boldsymbol{y}) + 0.5 * WL_{PC}(\boldsymbol{x}, \boldsymbol{y}), \quad (8)$$

where $MSL_{PC}(\boldsymbol{x}, \boldsymbol{y})$ and $WL_{PC}(\boldsymbol{x}, \boldsymbol{y})$ are calculated based on Equations 7 and 4, respectively.

## 4 Experiments

In our experiments, we utilized six DNNs, namely ResNet18 and ResNet50 He et al. (2016), WideResNet50 Zagoruyko & Komodakis (2016), ResNeXt50 Xie et al. (2017), VGG-16 and VGG-19 Simonyan & Zisserman (2015). Furthermore, we evaluate our two novel variants i.e., WPCL and MSPCL, and their combination on two automated driving datasets including two traffic sign classification datasets Stallkamp et al. (2011) and Temel et al. (2017). Moreover, we also used various OOD methods i.e., two variants including Isolation Forest (IF) Liu et al. (2008), Virtual Logit Matching (ViM) Wang et al. (2022), KL Matching Hendrycks et al. (2022), Max Softmax Probability (MSP) Hendrycks & Gimpel (2017), Energy-based methods Liu et al. (2020), ODINLiang et al. (2018), ReAct Sun et al. (2021), Maximum Logit Value (MaxLogit) Hendrycks et al. (2022) and k-NN based approaches Sun et al. (2022).

In the following, we explain our experiments per each category. This includes our training setup for the DNNs and the datasets, as well as our grouping of similar classes needed for our WPCL and MSPCL approaches. Followed by that, we explain the adversarial and non-adversarial augmentation methods used for generating OOD data on the GTSRB dataset. Finally, we explain the probabilistic methods used for OOD detection at the end of this section.

### 4.1 Datasets

We used the German Traffic Sign Recognition Benchmark (GTSRB) dataset Stallkamp et al. (2011) for training the mentioned DNNs that includes $50,000$ images with 43 classes. The image sizes used for training are $299 \times 299$ pixels. This dataset is then grouped into five different groups as well as four single classes. The groups include 1) speed limit signs with 9 classes; 2) triangular warning signs with 15 classes; 3) round blue direction signs with 8 classes, 4) no passing, no passing trucks, no trucks, and no vehicles; 5) end of all speed and passing limits, end of passing limit and end of passing truck limit. Individual classes include yield, stop, no entry, and right of way.

Furthermore, we extended our experiments to the CURE-TSR real dataset, which includes 49 video sequences with 14 classes. We followed the same image size for this dataset as the GTSRB. The groups in this dataset

include 1) goods vehicles and no overtaking; 2) no stopping and no parking; and 3) no left, no right, and priority. Individual classes include speed limits, stop, bicycle, hump, no entry, yield, and parking.

## 4.2 Training Parameters

As mentioned earlier, we have trained four DNNs including the ResNet18 and ResNet50 He et al. (2016), WideResNet50 Zagoruyko & Komodakis (2016), ResNeXt50 Xie et al. (2017), and VGG16 and VGG19 Simonyan & Zisserman (2015). Each of these DNNs is trained four times including with the conventional cross-entropy loss (CE), the prototype conformity loss (PCL) Mustafa et al. (2021), our WPCL, and our MSPCL. For all these types of training, we have used the ADAM optimizer with a learning rate of $1e-4$. Furthermore, for optimizing the p-norm balls in all the PCL-based training, we have used the stochastic gradient descent SGD optimizer with a learning rate of $5e-1$ for the proximity optimizer and $1e-4$ for the contrastive proximity optimizer.

## 4.3 OOD Data

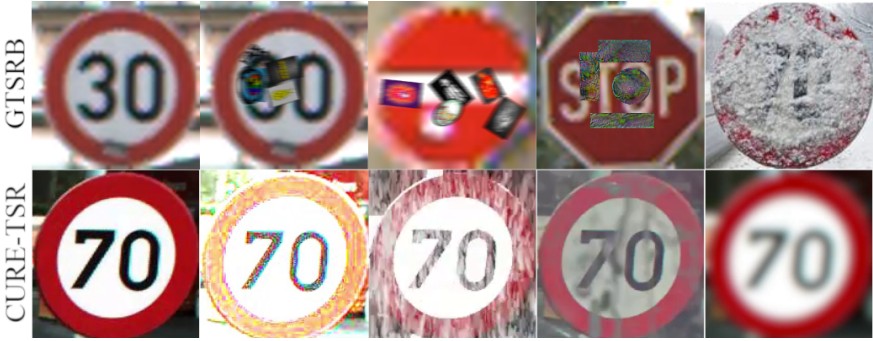

Figure 4: Examples of different OOD samples from both of the GTSRB Stallkamp et al. (2011) and the CURE-TSR Temel et al. (2017) datasets. The OOD data based on the GTSRB dataset is generated as part of the experiments conducted for this paper, while the OOD data based on the CURE-TSR dataset are part of the dataset itself.

We applied different augmentation methods to the GTSRB dataset to generate OOD data. These OOD datasets were then used to test the OOD detection performance. Based on that, we applied the RP2 Eykholt et al. (2017) and AP Brown et al. (2018) from the literature for attacking the image classifiers above. The RP2 attacks are optimized on five classes out of the 43 classes present in the GTSRB dataset, which include 30km/h, yield, STOP sign, no entry, and children crossing. Besides the two adversarial methods above, we applied the MCSA from Bayzidi et al. (2022) to the mentioned five attacked classes. As all the aforementioned augmentation methods introduced occlusion patterns to the input images, we also applied the well-known projected gradient descent (PGD) Madry et al. (2018) attack from the literature, which is an adversarial imperceptible attack. Moreover, we collected 261 images from the internet as real OOD data which include classes that are not presented in the dataset, heavily occluded images due to vandalism, snow, dirt, *etc.* .

For the CURE-TSR dataset, there are 12 augmentations per video sequence with five severity levels to reproduce real-world conditions that might challenge the traffic sign detectors, which are introduced by the authors. There we did not apply any further augmentation methods or adversarial attacks to this dataset and used the available augmentations. However, such OOD data are not used during the training of the DNNs and are solely used for testing the OOD detection methods during the test phase.

Examples of different augmentation methods leading to OOD data samples are illustrated in Figure 4, which include examples generated based on the GTSRB dataset (top row) and examples from the CURE-TSR dataset (bottom row). The first column includes clean images, while the rest are OOD data. These GTSRB examples in columns two to four are all generated for the ResNet18 DNN. The last column on the top row includes a real-world traffic sign covered with snow.

### 4.4 OOD Detection Benchmark

We evaluate all trained DNNs using the publicly available benchmark PyTorch-OOD Kirchheim et al. (2022). The complete benchmark consists Mahalanobis Lee et al. (2018), ViM Wang et al. (2022), ReAct Sun et al. (2021), KNN Sun et al. (2022), and IsolationForest Liu et al. (2008). We employ the same parameters for every evaluation, setting $\epsilon = 0.002$ for Mahalanobis, the projection dimension of $d = 64$ for ViM, and $k = 1000$ classes for KNN. Feature-based approaches are fit on the penultimate layer activations for all architectures. For our benchmark, we report two widely used metrics in the OOD detection domain. One is the AUROC which is the area under the ROC curve, describing the relationship of the TPR and FPR. Here, the higher the better. In addition, we report the FPR@95% metric measuring the false positive rate at a true positive rate threshold of 95%. Here, lower values are better.

## 5 Results and Discussion

In this section, we review the results of our experiments along with our observations, which are conducted over multiple DNNs and adversarial and non-adversarial augmentations to generate OOD data. The training methods include the conventional cross-entropy loss (CE), the original prototype conformity loss (PCL) Mustafa et al. (2021), our weighted version of the PCL method (WPCL), and our multi-scale version of the PCL method (MSPCL), and their combination (COM). The augmentation methods include the adversarial patch (AP) Brown et al. (2018), PGD Madry et al. (2018), RP2 Eykholt et al. (2017) and MCSA Bayzidi et al. (2022). The results are averaged over all the classes regardless of whether the OOD data samples inferred into those DNNs are classified correctly or not. The reason for such a decision is that although the OOD data might be classified correctly by the DNN, it is still OOD data and, therefore, needs to be classified as OOD as well.

We first show the classification results on GTSRB and CURE-TSR datasets in Table 5. Given GTSRB dataset, on average, CE outperforms its counterparts PCL, WPCL, and MSPCL in F1-score by a very small margin of 0.19%, 0.15% and 0.72% and in accuracy by 0.19%, 0.09% and 0.46% respectively. However, on CURE-TSR dataset, our proposed PCL variant, i.e., WPCL outperforms CE, PCL, and MSPCL in F1-score by a significant margins of 3.36%, 1.63% and 1.97% but in accuracy, PCL exceeds the performance of CE, WPCL, and MSPCL by a subtle margins of 0.22%, 0.17% and 20.97% respectively. Furthermore, we would like to emphasize that our goal in this paper is not to increase the classification performance or the robustness of the studied DNNs against the OOD data. On the other hand, by reporting these results, one can conclude that, except for certain DNNs, our proposed training approaches not only do not lead to a significant classification performance drop but also lead to an increase in classification performance for certain DNNs, such as ResNet18 and ResNet50 and datasets, such as the CURE-TSR. Examples of noticeable drops in performance can be observed in specific training setups, such as VGG16 and ResNeXt50 on the CURE-TSR dataset. This aligns with our motivation to introduce our training methods, which encourage the dispersed projection of latent activations for similar classes, which are not as well annotated in CURE-TSR compared to GTSRB. In fact, instead of precise annotation of classes such as speed limit signs, they are all grouped into singular classes, which then neutralizes the effectiveness of our approach.

The overall results of the OOD detection on the GTSRB dataset are presented in Table 2. The results are presented based on each DNN optimized four times with the different optimization methods (i.e., CE, PCL, WPCL, MSPCL) and tested with six of the OOD detection methods (IF, class-based IF, KNN, Mahalanobis distance, ReAct, and ViM). The reported metrics are AUROC and false positive rate. This table's last two columns represent each row's average results per metric. According to this table, it is observable that optimizing the DNNs with different versions of PCL leads to better OOD detection compared to conventional cross-entropy training.

It can be observed that except for the ResNet18 and ResNet50, all of the other tested DNNs achieved better OOD detection results on both of the reported metrics while they were optimized using either of our proposed PCL variations. Based on that, averaged over all the tested DNNs and OOD detection methods, the optimization with CE loss achieved 55.29 in AUROC and 79.36 in FPR. Moreover, the optimization with the original PCL achieved 55.45 and 75.21 on both of the metrics, respectively. However, with our proposed

| Model | | GTSRB | | | | CURE-TSR | | | |
|---|---|---|---|---|---|---|---|---|---|
| | | Accuracy | Precision | Recall | F1-Score | Accuracy | Precision | Recall | F1-Score |
| ResNet18 | **CE** | 99.26 | 98.99 | 99.32 | 99.12 | 95.74 | 74.81 | 81.53 | 75.44 |
| | **PCL** | 99.17 | 98.96 | 99.02 | 98.96 | 97.21 | 74.77 | 75.47 | 75.06 |
| | **WPCL** | 99.35 | 99.15 | 99.32 | 99.22 | 97.09 | 81.08 | 81.36 | 81.15 |
| | **MSPCL** | 99.52 | 99.37 | 99.50 | 99.43 | 94.12 | 78.83 | 81.14 | 79.73 |
| | **COM** | 99.03 | 98.72 | 98.91 | 98.77 | - | - | - | - |
| ResNet50 | **CE** | 99.02 | 97.95 | 98.59 | 98.13 | 95.77 | 68.64 | 68.12 | 68.30 |
| | **PCL** | 98.97 | 97.68 | 98.66 | 97.89 | 95.95 | 80.00 | 85.64 | 78.94 |
| | **WPCL** | 99.18 | 98.21 | 98.97 | 98.39 | 94.09 | 71.97 | 74.19 | 72.76 |
| | **MSPCL** | 98.76 | 97.78 | 98.78 | 98.14 | 96.49 | 74.30 | 75.20 | 74.68 |
| | **COM** | 98.98 | 98.62 | 98.33 | 98.38 | - | - | - | - |
| WideResNet50 | **CE** | 99.28 | 98.89 | 99.05 | 98.90 | 95.02 | 69.26 | 74.84 | 69.49 |
| | **PCL** | 98.67 | 98.40 | 97.83 | 98.02 | 95.47 | 73.57 | 74.68 | 74.02 |
| | **WPCL** | 99.12 | 98.80 | 98.89 | 98.81 | 96.04 | 80.66 | 81.46 | 80.99 |
| | **MSPCL** | 98.92 | 97.46 | 98.46 | 97.62 | 95.17 | 79.55 | 80.75 | 79.94 |
| | **COM** | 98.06 | 96.99 | 97.39 | 96.90 | - | - | - | - |
| ResNeXt50 | **CE** | 99.06 | 99.01 | 98.60 | 98.74 | 96.88 | 81.57 | 82.20 | 81.83 |
| | **PCL** | 99.27 | 99.00 | 98.95 | 98.94 | 97.72 | 83.84 | 90.16 | 85.47 |
| | **WPCL** | 99.06 | 98.14 | 98.98 | 98.40 | 96.97 | 79.58 | 82.73 | 80.88 |
| | **MSPCL** | 99.11 | 98.91 | 98.45 | 98.65 | 96.97 | 76.78 | 76.72 | 76.67 |
| | **COM** | 98.94 | 98.49 | 98.03 | 98.16 | - | - | - | - |
| VGG16 | **CE** | 98.99 | 97.51 | 98.54 | 97.74 | 95.11 | 73.53 | 74.37 | 73.80 |
| | **PCL** | 98.69 | 97.38 | 98.62 | 97.78 | 93.79 | 72.41 | 72.09 | 72.07 |
| | **WPCL** | 98.90 | 97.87 | 98.13 | 97.82 | 94.51 | 68.09 | 68.55 | 68.22 |
| | **MSPCL** | 97.59 | 96.93 | 96.66 | 96.66 | 92.50 | 64.67 | 68.95 | 66.52 |
| | **COM** | 98.46 | 98.46 | 98.19 | 98.29 | - | - | - | - |
| VGG19 | **CE** | 99.21 | 98.76 | 98.78 | 98.71 | 94.42 | 72.74 | 73.87 | 73.18 |
| | **PCL** | 98.95 | 98.89 | 98.48 | 98.63 | 94.12 | 67.71 | 66.41 | 66.91 |
| | **WPCL** | 98.69 | 98.39 | 97.48 | 97.80 | 94.54 | 78.23 | 78.60 | 78.22 |
| | **MSPCL** | 98.19 | 96.76 | 96.70 | 96.52 | 93.37 | 72.98 | 72.97 | 72.87 |
| | **COM** | 98.03 | 98.23 | 96.92 | 97.44 | - | - | - | - |
| **Average** | **CE** | **99.14** | **98.52** | **98.81** | **98.56** | 95.49 | 73.43 | 75.82 | 73.68 |
| | **PCL** | 98.95 | 98.39 | 98.59 | 98.37 | **95.71** | 75.38 | 77.41 | 75.41 |
| | **WPCL** | 99.05 | 98.43 | 98.63 | 98.41 | 95.54 | **76.60** | **77.82** | **77.04** |
| | **MSPCL** | 98.68 | 97.87 | 98.09 | 97.84 | 94.77 | 74.52 | 75.96 | 75.07 |
| | **COM** | 98.58 | 98.25 | 97.96 | 97.99 | - | - | - | - |

Table 1: Training classification performance results of all the six DNNs with all four training methods on both of the clean test datasets. The trained DNNs include ResNet18 and ResNet50 He et al. (2016), WideResNet50 Zagoruyko & Komodakis (2016), ResNeXt50 Xie et al. (2017), and the VGG16 and VGG19 Simonyan & Zisserman (2015). The training methods include training with the conventional cross-entropy (CE) loss, the PCL Mustafa et al. (2021), our WPCL, and our MSPCL. The performance evaluation metrics include accuracy, precision, recall, and F1-score.

| Model | | IF | | KNN | | Mahal. | | ReAct | | ViM | | Average | |
|---|---|---|---|---|---|---|---|---|---|---|---|---|---|
| | | AUROC↑ | FPR↓ | AUROC↑ | FPR↓ | AUROC↑ | FPR↓ | AUROC↑ | FPR↓ | AUROC↑ | FPR↓ | AUROC↑ | FPR↓ |
| ResNet18 | CE | 63.66 | 86.28 | 68.16 | 67.32 | 34.99 | 97.95 | 68.41 | 75.97 | 69.26 | 75.89 | 60.9 | 80.68 |
| | PCL | 73.63 | 57.28 | 39.49 | 79.53 | 33.99 | 88.02 | 85.77 | 43.25 | 51.56 | 65.14 | 56.89 | 66.64 |
| | WPCL | 60.31 | 98.63 | 61.91 | 67.62 | 48.56 | 75.95 | 58.45 | 79.66 | 57.94 | 82.12 | 57.43 | 80.80 |
| | MSPCL | 70.21 | 76.01 | 59.43 | 82.3 | 27.49 | 96.19 | 56.43 | 87.0 | 66.37 | 78.92 | 55.99 | 84.08 |
| | COM. | 70.9 | 62.25 | 48.6 | 74.79 | 34.71 | 80.68 | 83.14 | 52.48 | 73.74 | 55.07 | **62.22** | **65.05** |
| ResNet50 | CE | 55.69 | 88.95 | 55.96 | 85.14 | 44.31 | 93.74 | 59.04 | 77.69 | 65.84 | 66.42 | **56.17** | 82.39 |
| | PCL | 61.5 | 81.87 | 39.05 | 93.79 | 45.11 | 91.23 | 70.06 | 76.55 | 58.89 | 81.12 | 54.92 | 84.91 |
| | WPCL | 45.69 | 93.45 | 44.84 | 94.69 | 19.5 | 64.59 | 59.49 | 80.27 | 46.78 | 88.51 | 43.26 | 84.30 |
| | MSPCL | 74.99 | 61.21 | 53.66 | 91.25 | 21.23 | 99.85 | 49.82 | 76.65 | 51.98 | 74.5 | 50.34 | 80.69 |
| | COM. | 64.23 | 71.2 | 49.48 | 75.45 | 2.94 | 97.07 | 57.39 | 87.59 | 61.45 | 71.65 | 47.1 | **80.59** |
| WideResNet50 | CE | 58.77 | 86.81 | 59.7 | 79.96 | 38.26 | 94.29 | 61.16 | 78.52 | 66.89 | 70.89 | 56.96 | 82.09 |
| | PCL | 51.73 | 80.05 | 36.76 | 74.92 | 6.71 | 91.06 | 55.03 | 75.08 | 58.69 | 77.04 | 41.78 | 79.63 |
| | WPCL | 61.08 | 93.4 | 52.81 | 89.57 | 43.09 | 98.53 | 49.87 | 88.46 | 49.5 | 90.99 | 51.27 | 92.19 |
| | MSPCL | 74.77 | 67.6 | 66.09 | 79.93 | 44.85 | 85.82 | 49.8 | 79.47 | 61.00 | 74.40 | **59.30** | **77.44** |
| | COM. | 67.71 | 73.83 | 56.32 | 85.41 | 17.37 | 96.44 | 61.43 | 85.59 | 66.04 | 72.49 | 53.77 | 82.75 |
| ResNeXt50 | CE | 66.88 | 81.43 | 63.97 | 79.77 | 3.16 | 99.82 | 62.06 | 84.35 | 26.96 | 99.52 | 44.61 | 88.98 |
| | PCL | 58.94 | 82.32 | 47.3 | 78.5 | 58.55 | 80.8 | 74.62 | 69.96 | 52.73 | 91.6 | **58.43** | 80.64 |
| | WPCL | 43.62 | 92.01 | 48.22 | 93.84 | 52.79 | 96.05 | 49.26 | 88.8 | 44.82 | 94.94 | 47.74 | 93.13 |
| | MSPCL | 81.7 | 56.15 | 67.77 | 80.37 | 28.38 | 98.02 | 49.82 | 76.89 | 50.15 | 77.86 | 55.56 | **77.86** |
| | COM. | 67.97 | 73.63 | 48.84 | 76.21 | 14.63 | 97.32 | 69.72 | 85.93 | 67.46 | 75.26 | 53.72 | 81.67 |
| VGG16 | CE | 70.32 | 55.61 | 34.95 | 86.23 | 37.76 | 78.29 | 82.13 | 56.33 | 81.03 | 72.75 | 61.24 | 69.84 |
| | PCL | 70.53 | 55.44 | 34.3 | 83.26 | 61.69 | 62.41 | 83.94 | 51.93 | 80.65 | 69.85 | 62.27 | 64.58 |
| | WPCL | 70.35 | 51.7 | 44.52 | 79.22 | 40.78 | 78.92 | 84.59 | 54.59 | 69.61 | 56.83 | 61.97 | 64.26 |
| | MSPCL | 72.44 | 50.23 | 44.39 | 81.3 | 30.86 | 79.33 | 79.99 | 44.2 | 89.68 | 46.18 | **63.47** | **60.25** |
| | COM. | 70.43 | 60.59 | 36.47 | 82.27 | 21.69 | 83.99 | 81.8 | 49.37 | 71.83 | 61.24 | 56.44 | 67.49 |
| VGG19 | CE | 69.4 | 58.55 | 37.49 | 80.18 | 34.88 | 79.72 | 50.29 | 75.19 | 67.15 | 67.2 | 51.84 | 72.17 |
| | PCL | 66.04 | 74.25 | 35.1 | 80.79 | 76.39 | 50.24 | 48.23 | 77.53 | 46.77 | 91.5 | 54.51 | 74.86 |
| | WPCL | 75.15 | 48.61 | 35.57 | 77.15 | 68.02 | 62.86 | 83.15 | 52.07 | 78.56 | 52.91 | **68.09** | **58.72** |
| | MSPCL | 70.29 | 67.36 | 34.19 | 84.74 | 70.78 | 58.92 | 86.26 | 52.65 | 74.81 | 61.26 | 67.27 | 64.99 |
| | COM. | 71.11 | 53.77 | 38.17 | 92.45 | 22.54 | 84.68 | 77.79 | 58.54 | 68.99 | 73.2 | 55.72 | 72.53 |
| Average | CE | 64.12 | 76.27 | 53.37 | **79.77** | 32.23 | 90.63 | 63.85 | 74.67 | 62.85 | 75.45 | 55.29 | 79.36 |
| | PCL | 63.73 | 71.87 | 38.67 | 81.8 | **47.07** | **77.29** | 69.61 | **65.72** | 58.22 | 79.38 | 55.45 | 75.21 |
| | WPCL | 59.37 | 79.63 | 47.98 | 83.68 | 45.46 | 79.47 | 64.14 | 73.97 | 57.87 | 77.72 | 54.96 | 75.65 |
| | MSPCL | **74.07** | **63.09** | **54.25** | 83.31 | 37.27 | 86.35 | 62.02 | 69.48 | 65.67 | 68.85 | **58.65** | **74.21** |
| | COM. | 68.73 | 65.88 | 46.31 | 81.1 | 18.98 | 90.03 | **71.88** | 69.92 | **68.25** | **68.15** | 54.83 | 75.01 |

Table 2: The OOD detection results on the GTSRB dataset, the six trained DNNs, four training methods, and the six OOD detection methods. The OOD data used for this evaluation is generated with five augmentation methods and a dataset consisting of 260 OOD images collected from the internet. The optimization methods include 1) the common cross-entropy (CE) loss; 2) the original PCL; 3) our WPCL; 4) and our MSPCL. The OOD detection methods include IF, class-based IF, KNN, Mahalanobis distance, ReAct, and ViM. The reported metrics are AUROC and false positive rate at 95 percept true positive rate. The last two columns on the right indicate the average over all the OOD detection methods on each row and each metric. Please note that the OOD detection methods results are averaged over 5 for different random seeds used for optimizing the OOD detection methods.

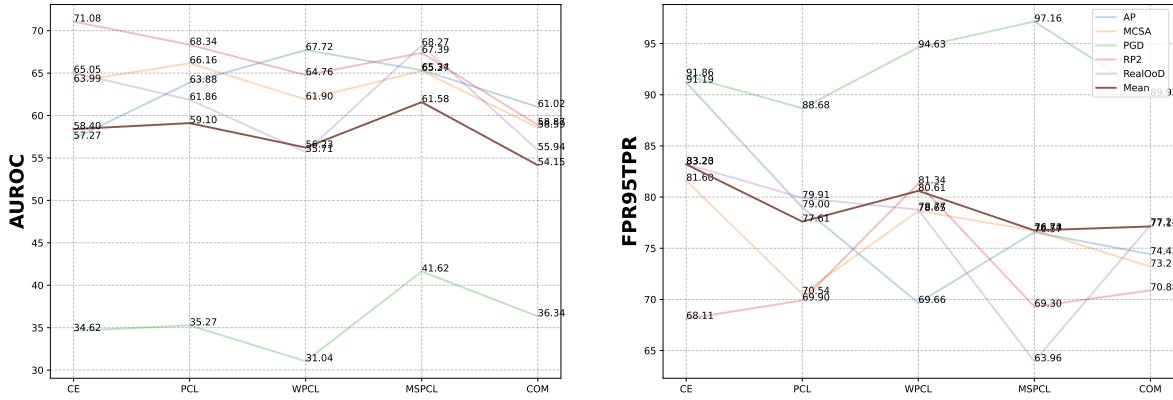

Figure 5: OOD detection results across different training methods and OOD data types.

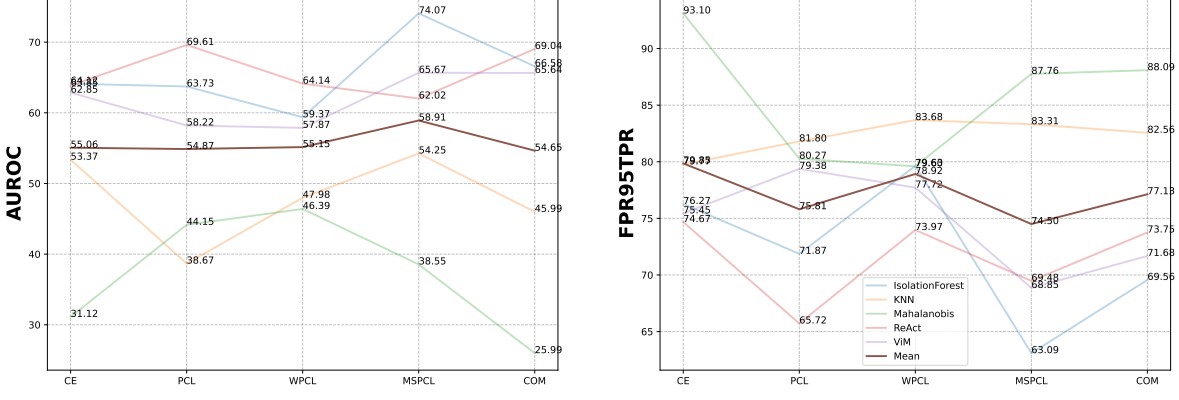

Figure 6: OOD detection results across different training and OOD detection methods.

novel variants of PCL, i.e., WPCL and MSPCL, we comprehensively outperformed both the CE and PCL with MSPCL on average. For WPCL, we achieved an AUROC of 54.96 and an FPR of 75.65. Furthermore, for our MSPCL, we achieved an AUROC of 58.65 and an FPR of 74.21. Accordingly, compared to all the other DNNs, the VGG16 has benefitted the most from our proposed training methodology, achieving a boost of 16.25 in AUROC and 13.45 in FPR. This indicates that in use cases such as traffic sign classification, which includes a variety of similar classes, disentangling them using our proposed methods would be a more promising approach to achieve better outlier detection.

The Figures 7 and 8 depict the predicted scores of an IsolationForest from our detection benchmark on the clean versus RP2 attacked, as well as clean versus real OOD test samples. The IsolationForest was fitted on a ResNet50 trained on CE, PCL, and our WPCL and MSPCL. It can be observed that the predicted scores have a significant overlap for the CE-trained network, which makes distinguishing inliers and outliers difficult. One can also observe that defining a threshold to separate in-distribution and OOD would only be effectively possible for MSPCL due to its strong separability of clean and OOD data. These results are also in coordination with the detailed results presented in Tables S.2 and S.1 in Supplementary Section A.

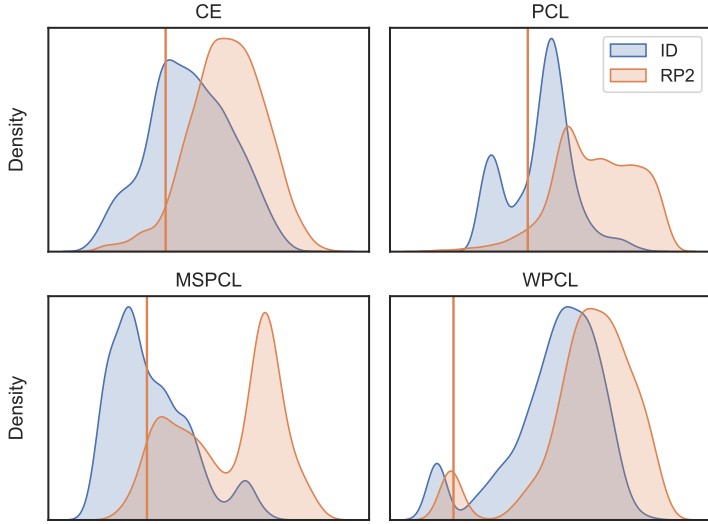

Figure 7: Distributions of predicted scores on the clean test samples and RP2 attacked test samples by Isolation Forest fitted on a ResNet50 He et al. (2016) trained with CE, PCL Mustafa et al. (2021), our WPCL and MSPCL. The vertical line indicates the threshold, whereby 95 percent of the OOD data would be selected as OOD.

Figures 5 and 6 illustrate the results across different training methods, OOD data types, and OOD detection methods. One can observe from these plots that different OOD data types including different adversarial attacks, real-world OOD data, and other augmentations, can achieve different results across different training approaches. However, one can observe that except for RP2, in all the other OOD types, either of the PCL variants outperforms the OOD detection when the DNNs are trained with cross-entropy loss. This indicates a clear conclusion that training with such an approach is beneficial for OOD detection. On the other hand, one can also observe that all the OOD detection methods benefited from either variant of the PCL training. However, different results might also indicate inconsistency in selecting the best method for this task. Therefore, it is important to note that each DNN architecture might benefit differently from such training approaches, and one would need to perform extensive analysis for each architecture to select the best configuration.

As mentioned earlier, we have also extended our experiments to the CURE-TSR dataset, which includes only 14 classes. This led to the definition of smaller groups for both the WPCL and MSPCL. Therefore, the results of the different OOD detection performances on different training approaches on the CURE-TSR dataset are presented in Table 3. The results are presented based on each DNN optimized four times with the different optimization methods (i.e. CE, PCL, WPCL, MSPCL) and tested with five of the OOD detection methods (IF, KNN, Mahalanobis distance, ReAct, and ViM). The reported metrics are AUROC and false positive rate. The last two columns of this table represent the average results of each row per metric. Moreover, as the dataset includes 12 augmentation with 5 severity levels each, we observed that up to severity level 3, the augmentations did not lead to a major drop in the classification performance of the networks. In fact, levels 4 and 5 led to a more than 20 percent decrease in the classification performance of the evaluated DNNs on average. Therefore, we considered the data samples of these two levels as OOD data and evaluated the benchmarked OOD detection methods only on these two levels. Accordingly, the results presented in Table 3 include only these two levels. According to these results, our proposed approaches can outperform the other evaluated ones in some of the DNNs, such as VGG19, WideResNet50 (only on AUROC), and ResNet18. However, it can be observed that OOD detection methods such as Mahalanobis distance benefit greatly from our proposed approaches, in particular the WPCL.

We performed a dispersion analysis on the activations generated from the WideResNet50 DNN trained on the GTSRB dataset. The results of this analysis are presented in Figure 9. In order to quantify how far the

| Model | | IF | | KNN | | Mahal. | | ReAct | | ViM | | Average | |
|---|---|---|---|---|---|---|---|---|---|---|---|---|---|
| | | **AUROC↑** | **FPR↓** | **AUROC↑** | **FPR↓** | **AUROC↑** | **FPR↓** | **AUROC↑** | **FPR↓** | **AUROC↑** | **FPR↓** | **AUROC↑** | **FPR↓** |
| ResNet18 | **CE** | 53.44 | 94.19 | 49.17 | 91.38 | 60.7 | 76.32 | 64.51 | 84.82 | 68.93 | 74.85 | 59.35 | 84.31 |
| | **PCL** | 35.54 | 95.22 | 53.66 | 87.64 | 70.24 | 66.58 | 66.03 | 87.56 | 72.67 | 76.08 | 59.63 | 82.62 |
| | **WPCL** | 38.82 | 90.76 | 53.88 | 86.19 | 70 | 67.64 | 65.79 | 86.73 | 71.5 | 78.24 | 60.0 | 81.91 |
| | **MSPCL** | 60.21 | 87.86 | 64.29 | 83.46 | 63.94 | 73.72 | 65.32 | 85.39 | 69.66 | 78.84 | **64.68** | **81.85** |
| ResNet50 | **CE** | 55.43 | 92.68 | 53.64 | 92.7 | 55.89 | 63.06 | 65.05 | 83.7 | 71.08 | 74.95 | **60.22** | **81.42** |
| | **PCL** | 39.82 | 93.95 | 51.33 | 88.06 | 69.1 | 63.65 | 63.3 | 90.61 | 68.49 | 84.65 | 58.41 | 84.18 |
| | **WPCL** | 40.75 | 92.87 | 51.4 | 89.8 | 69.51 | 65.65 | 61.58 | 91.07 | 59.56 | 91.42 | 56.56 | 86.16 |
| | **MSPCL** | 48.96 | 91.95 | 56.2 | 90.77 | 43 | 98.4 | 60.09 | 90.86 | 62.33 | 90.51 | 54.12 | 92.5 |
| WideResNet50 | **CE** | 57.87 | 91.03 | 53.76 | 90.35 | 17.09 | 73.22 | 64.3 | 64.3 | 67.63 | 80.34 | 52.13 | 83.67 |
| | **PCL** | 44.14 | 92.95 | 51.27 | 86.98 | 60.58 | 67.63 | 67.7 | 86.92 | 69.31 | 83.66 | 58.56 | 84.23 |
| | **WPCL** | 42.49 | 94.08 | 49.45 | 91.47 | 67.27 | 70.3 | 61.85 | 91.59 | 52.46 | 95.63 | **58.6** | **83.63** |
| | **MSPCL** | 57.82 | 89.94 | 62.7 | 84.85 | 45.95 | 72.04 | 64.44 | 84.94 | 61.92 | 89.36 | 54.7 | 88.61 |
| ResNeXt50 | **CE** | 55.41 | 91.22 | 52.61 | 90.72 | 42.75 | 71.22 | 64.93 | 82.29 | 68.78 | 76.87 | 56.9 | **82.46** |
| | **PCL** | 40.22 | 93.3 | 53.21 | 88.72 | 65.41 | 67.34 | 64.59 | 89.98 | 68.62 | 80.12 | **58.41** | 83.89 |
| | **WPCL** | 37.9 | 96.65 | 48 | 92.59 | 67.65 | 68.37 | 56.22 | 91.49 | 35.74 | 95.61 | 49.1 | 88.94 |
| | **MSPCL** | 39.26 | 94.12 | 50.2 | 91.46 | 66.18 | 71.6 | 62.38 | 90.62 | 50 | 94.88 | 53.6 | 88.54 |
| VGG16 | **CE** | 62.95 | 80.33 | 52.97 | 86.45 | 36.61 | 97.38 | 69.61 | 77.05 | 64.71 | 79.95 | 57.37 | 84.23 |
| | **PCL** | 58.96 | 84.11 | 50.79 | 82.68 | 54.92 | 90.35 | 75.05 | 71.47 | 73.49 | 73.64 | **62.64** | **80.45** |
| | **WPCL** | 54.09 | 89 | 51.64 | 86.98 | 61.3 | 85.25 | 71.94 | 72.06 | 71.99 | 70.17 | 62.19 | 80.69 |
| | **MSPCL** | 53.45 | 86.38 | 46.3 | 90.39 | 63.42 | 82.56 | 70.4 | 76.07 | 71.5 | 74.04 | 61.01 | 81.89 |
| VGG19 | **CE** | 65.03 | 78.24 | 50.74 | 85.31 | 36.3 | 97.81 | 70.15 | 72.3 | 65.48 | 76.39 | 57.54 | 82.01 |
| | **PCL** | 57.16 | 84 | 53.72 | 79.78 | 61.3 | 89.31 | 75.2 | 70.75 | 74.26 | 72.34 | 64.33 | 79.23 |
| | **WPCL** | 53.44 | 87.24 | 55.27 | 76.33 | 68.11 | 79.03 | 77.98 | 60.69 | 77.33 | 62.01 | **66.43** | **73.06** |
| | **MSPCL** | 48.44 | 91.91 | 49.12 | 87.5 | 53.84 | 90.01 | 71.32 | 73.51 | 69.07 | 75.36 | 58.36 | 83.66 |
| Average | **CE** | **58.35** | **87.95** | 52.15 | 89.48 | 41.56 | 79.83 | 66.42 | **80.59** | 67.77 | **77.22** | 57.25 | 83.02 |
| | **PCL** | 45.97 | 90.59 | 52.33 | **85.64** | 63.59 | 74.14 | **68.65** | 82.88 | **71.14** | 78.41 | **60.34** | **82.33** |
| | **WPCL** | 44.58 | 91.77 | 51.61 | 87.23 | **67.31** | **72.7** | 65.89 | 82.27 | 61.43 | 82.18 | 58.16 | 83.23 |
| | **MSPCL** | 51.35 | 90.36 | **54.8** | 88.07 | 56.06 | 81.39 | 65.66 | 83.56 | 64.08 | 83.83 | 58.39 | 85.44 |

Table 3: The OOD detection results on the CURE-TSR dataset, the six trained DNNs, four training methods, and the six OOD detection methods. The OOD data used for this evaluation is generated with twelve augmentation methods with five severity levels included in the dataset, out of which severity levels four and five are considered OOD, and used for this evaluation. The optimization of the DNNs is done on only clean data. The optimization methods include 1) the common cross-entropy (CE) loss; 2) the original PCL; 3) our WPCL; 4) and our MSPCL. The OOD detection methods include IF, class-based IF, KNN, Mahalanobis distance, ReAct, and ViM. The reported metrics are AUROC and false positive rate at 95 percept true positive rate. The last two columns on the right indicate the average over all the OOD detection methods on each row and each metric.

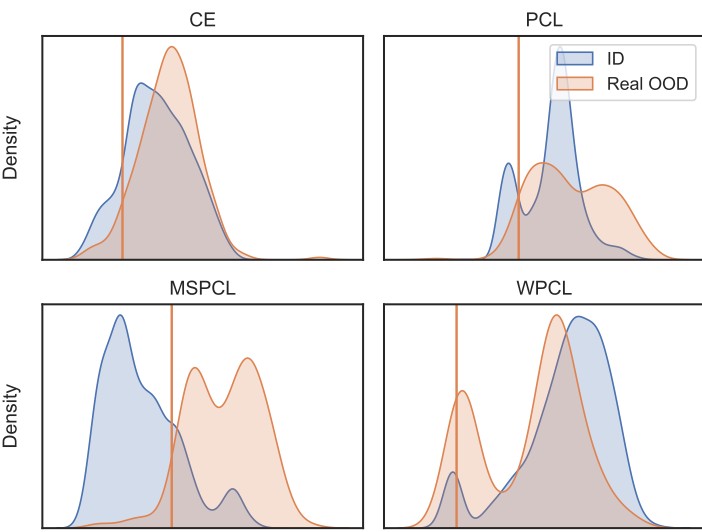

Figure 8: Distributions of predicted scores on the clean test samples and real OOD test samples by Isolation Forest fitted on a ResNet50 He et al. (2016) trained with CE, PCL Mustafa et al. (2021), our WPCL and MSPCL. The vertical line indicates the threshold, whereby 95 percent of the OOD data would be selected as OOD.

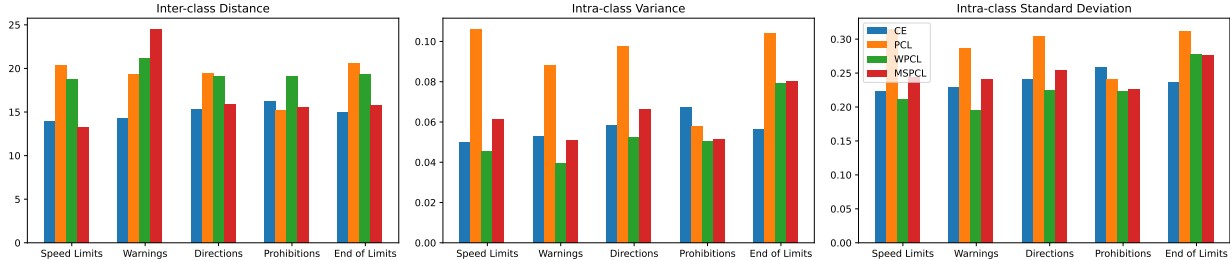

Figure 9: Dispersion analysis based on inter-class distance, intra-class variance, and intra-class standard deviation of the activations extracted from the WideResNet50 DNN trained on the GTSRB dataset. The results are calculated and averaged among the grouped classes.

similar classes within each group are projected, we calculated the distance of activations generated for each class with the other classes. To compare the compactness of each activation class, we measured the variance and standard deviation, which are averaged over each of the groups. As shown in this figure, our WPCL method achieved a bigger inter-class distance for all the groups while also achieving the lowest numbers in intra-class variance and standard deviation for most of the groups. This indicates that the WPCL could achieve more compact activations within classes while pushing them from other classes.

Despite the encouraging results showcasing the effectiveness of our proposed approaches in training the studied DNNs for traffic sign image classification, we consider the following limitations to be addressed as potential future work. While we only evaluated our methods on traffic sign image classification, we did not extend our studies to other image classification tasks that might benefit from this approach. Therefore, we propose to extend our methods to datasets containing multiple similar classes that could cause naive disentanglement by the encoder. Moreover, as our experiments are conducted on the ResNet, WideResNet, ResNeXt, and VGG DNNs, we propose to extend the experiments to other architectures to study their differences and similarities as well. Due to the extent of our experiments, we did not perform any fine-tuning of already trained DNNs, and therefore, we would encourage further research in this regard. On the other hand, we only utilized one contrastive training approach, PCL, to showcase the challenges and potential solutions when adopting SOTA solutions to real-world problems. Therefore, we also encourage further research with other contrastive or similar training approaches that can benefit from such an adaptation technique. Conducting experiments on different OOD methods, different attacks, and other augmentation techniques to generate OOD data would also enrich this work. To increase the DNN's robustness, we would encourage further research using our proposed approaches and adversarial training. Finally, we would like to emphasize that our work showcases an example solution for closing the gap between SOTA machine learning and the safety assurance of such methods in safety-critical applications, wherein active monitoring of all the components is essential to ensure safe operation in all scenarios including unforeseen malfunction of such components. The authors, therefore, highly encourage further research in this area.

## 6 Conclusion

In this paper, we discuss the problem of OOD detection for traffic sign image classification methods. This refers to the problem of CNN-based encoders where the projection of data samples of different classes lie very close to each other in latent space when optimized solely with CE loss function, this leads to poor OOD detection results based on CNN's latent activations. Therefore, we first proposed to apply the prototype conformity loss (PCL) function from the literature to the traffic sign classification problem. However, due to very similar groups in the dataset e.g., 30km/h and 80km/h, PCL failed to disentangle them properly. Therefore, to solve this problem, we introduced two novel variations of the PCL function, namely weighted prototype conformity loss WPCL and multi-scale prototype conformity loss MSPCL, to enhance the OOD detection rate for both clean data as well as OOD data. Finally, we have shown in our experiments that our proposed variants, MSPCL outperforms the CE and PCL by 3.36% and 3.20% in AUROC and

5.515% and 1.00% in FPR, across six DNN's and five OOD detection methods on the GTSRB dataset which includes various similar classes that can benefit from our proposed methodology. These results indicate that it is possible to introduce optimization goals besides the main classification task that are tailored to the need for monitoring DNNs within safety-critical applications during their runtime operations without disrupting their classification capability.

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

# A   Appendix

## Supplementary Information

In this section, the extra information and details about our experiments and findings are presented. Accordingly, the training details of all the DNNs and training methods are discussed in Section S.1. Following that, the extra results of two individual OOD datasets from the GTSRB experiments are presented in Section S.2. Finally, as different training methods lead to the different encoding of the image data samples, especially in latent spaces of the DNNs, we have visualized and discussed the activations for the WideResNet50 DNN in Section S.4, which include an attached video for better clarification.

### S.1   Training Details

As mentioned earlier, we have trained four DNNs including the ResNet18 and ResNet50 He et al. (2016), WideResNet50 Zagoruyko & Komodakis (2016), ResNeXt50 Xie et al. (2017), and VGG16 and VGG19 Simonyan & Zisserman (2015). Each of these DNNs is trained four times including with the conventional cross-entropy loss (CE), the prototype conformity loss (PCL) Mustafa et al. (2021), our WPCL, and our MSPCL. For all these types of training, we have used the ADAM optimizer with a learning rate of $1e-4$. Furthermore, for optimizing the p-norm balls in all the PCL-based training, we have used the stochastic gradient descent SGD optimizer with a learning rate of 0.5 for the proximity optimizer and $1e-4$ for the contrastive proximity optimizer.

| Model | | IF AUROC↑ | IF FPR↓ | KNN AUROC↑ | KNN FPR↓ | Mahal. AUROC↑ | Mahal. FPR↓ | ReAct AUROC↑ | ReAct FPR↓ | ViM AUROC↑ | ViM FPR↓ | Average AUROC↑ | Average FPR↓ |
|---|---|---|---|---|---|---|---|---|---|---|---|---|---|
| | CE | $73.41^{\pm0.0}$ | $78.87^{\pm0.0}$ | $79.93^{\pm0.0}$ | $63.38^{\pm0.0}$ | $29.56^{\pm0.0}$ | $95.77^{\pm0.0}$ | $79.43^{\pm0.34}$ | $84.71^{\pm2.46}$ | $80.76^{\pm0.36}$ | $76.1^{\pm1.72}$ | **68.61** | **79.77** |
| ResNet18 | PCL | $78.16^{\pm0.0}$ | $83.56^{\pm0.0}$ | $35.23^{\pm0.0}$ | $99.05^{\pm0.0}$ | $35.12^{\pm0.0}$ | $97.25^{\pm0.0}$ | $80.72^{\pm0.19}$ | $75.31^{\pm1.72}$ | $63.77^{\pm0.62}$ | $72.13^{\pm0.92}$ | 58.60 | 85.46 |
| | WPCL | $47.7^{\pm0.0}$ | $100.0^{\pm0.0}$ | $57.8^{\pm0.0}$ | $83.92^{\pm0.0}$ | $67.55^{\pm0.0}$ | $79.41^{\pm0.0}$ | $60.6^{\pm1.33}$ | $97.62^{\pm0.55}$ | $42.06^{\pm0.73}$ | $94.27^{\pm0.37}$ | 55.14 | 91.04 |
| | MSPCL | $74.78^{\pm0.0}$ | $71.53^{\pm0.0}$ | $50.45^{\pm0.0}$ | $90.05^{\pm0.0}$ | $37.56^{\pm0.0}$ | $96.35^{\pm0.0}$ | $46.18^{\pm0.56}$ | $96.93^{\pm0.53}$ | $59.12^{\pm0.26}$ | $90.95^{\pm0.58}$ | 53.60 | 89.16 |
| | CE | $62.07^{\pm0.0}$ | $87.07^{\pm0.0}$ | $70.93^{\pm0.0}$ | $86.08^{\pm0.0}$ | $37.58^{\pm0.0}$ | $95.5^{\pm0.0}$ | $69.68^{\pm0.26}$ | $74.94^{\pm0.48}$ | $78.2^{\pm0.34}$ | $59.09^{\pm1.19}$ | **62.94** | 81.03 |
| ResNet50 | PCL | $63.64^{\pm0.0}$ | $79.64^{\pm0.0}$ | $48.02^{\pm0.0}$ | $72.21^{\pm0.0}$ | $36.34^{\pm0.0}$ | $95.9^{\pm0.0}$ | $75.9^{\pm0.47}$ | $70.74^{\pm0.3}$ | $65.63^{\pm0.04}$ | $88.25^{\pm0.27}$ | 57.33 | 83.27 |
| | WPCL | $29.39^{\pm0.0}$ | $93.24^{\pm0.0}$ | $36.54^{\pm0.0}$ | $99.91^{\pm0.0}$ | $5.43^{\pm0.0}$ | $99.86^{\pm0.0}$ | $55.34^{\pm0.44}$ | $88.8^{\pm1.24}$ | $24.33^{\pm0.16}$ | $93.58^{\pm0.06}$ | 30.14 | 80.87 |
| | MSPCL | $93.09^{\pm0.0}$ | $21.58^{\pm0.0}$ | $71.83^{\pm0.0}$ | $69.14^{\pm0.0}$ | $0.36^{\pm0.0}$ | $99.86^{\pm0.0}$ | $50.0^{\pm0.0}$ | $76.97^{\pm1.41}$ | $55.34^{\pm0.19}$ | $47.7^{\pm1.51}$ | 54.67 | **64.2** |
| | CE | $69.99^{\pm0.0}$ | $84.05^{\pm0.0}$ | $71.04^{\pm0.0}$ | $76.22^{\pm0.0}$ | $31.23^{\pm0.0}$ | $91.89^{\pm0.0}$ | $73.52^{\pm0.3}$ | $79.43^{\pm0.68}$ | $78.97^{\pm0.3}$ | $77.33^{\pm1.14}$ | 64.95 | 81.78 |
| WideResNet50 | PCL | $50.98^{\pm0.0}$ | $76.49^{\pm0.0}$ | $39.54^{\pm0.0}$ | $69.1^{\pm0.0}$ | $74.31^{\pm0.03}$ | $86.95^{\pm0.38}$ | $58.29^{\pm0.69}$ | $71.17^{\pm0.46}$ | $59.53^{\pm0.5}$ | $72.59^{\pm0.71}$ | 48.53 | 75.58 |
| | WPCL | $55.61^{\pm0.0}$ | $93.24^{\pm0.0}$ | $44.11^{\pm0.0}$ | $97.97^{\pm0.0}$ | $45.53^{\pm0.02}$ | $99.91^{\pm0.0}$ | $50.0^{\pm0.0}$ | $95.89^{\pm1.79}$ | $47.91^{\pm0.14}$ | $92.66^{\pm0.1}$ | 48.63 | 95.93 |
| | MSPCL | $96.15^{\pm0.0}$ | $11.98^{\pm0.0}$ | $83.91^{\pm0.0}$ | $46.58^{\pm0.0}$ | $21.03^{\pm0.01}$ | $78.02^{\pm0.01}$ | $50.0^{\pm0.0}$ | $60.61^{\pm1.63}$ | $85.54^{\pm0.54}$ | $39.46^{\pm0.4}$ | **67.33** | **47.33** |
| | CE | $72.39^{\pm0.0}$ | $84.82^{\pm0.0}$ | $72.09^{\pm0.0}$ | $77.7^{\pm0.0}$ | $0.0^{\pm0.0}$ | $99.95^{\pm0.0}$ | $64.2^{\pm0.49}$ | $92.26^{\pm2.5}$ | $22.02^{\pm0.11}$ | $99.95^{\pm0.03}$ | 46.14 | 90.94 |
| ResNeXt50 | PCL | $45.54^{\pm0.0}$ | $99.86^{\pm0.0}$ | $47.47^{\pm0.0}$ | $83.24^{\pm0.0}$ | $74.34^{\pm0.0}$ | $89.46^{\pm0.0}$ | $77.34^{\pm0.5}$ | $73.77^{\pm1.43}$ | $48.92^{\pm0.23}$ | $93.24^{\pm0.0}$ | 58.72 | 87.91 |
| | WPCL | $23.09^{\pm0.0}$ | $92.21^{\pm0.0}$ | $27.86^{\pm0.0}$ | $99.23^{\pm0.0}$ | $47.01^{\pm0.0}$ | $96.71^{\pm0.0}$ | $50.07^{\pm0.09}$ | $96.4^{\pm0.58}$ | $28.86^{\pm0.07}$ | $97.06^{\pm0.23}$ | 35.38 | 96.32 |
| | MSPCL | $95.94^{\pm0.0}$ | $8.83^{\pm0.0}$ | $75.57^{\pm0.0}$ | $57.7^{\pm0.0}$ | $3.64^{\pm0.0}$ | $100.0^{\pm0.0}$ | $50.0^{\pm0.0}$ | $69.49^{\pm1.9}$ | $50.77^{\pm0.0}$ | $38.72^{\pm2.14}$ | **55.18** | **54.95** |
| | CE | $93.66^{\pm0.18}$ | $29.57^{\pm1.84}$ | $31.98^{\pm0.24}$ | $91.05^{\pm0.67}$ | $12.3^{\pm0.04}$ | $99.86^{\pm0.0}$ | $81.15^{\pm0.1}$ | $82.18^{\pm3.21}$ | $79.13^{\pm0.84}$ | $76.81^{\pm1.48}$ | 71.48 | 69.9 |
| VGG16 | PCL | $97.44^{\pm0.12}$ | $12.89^{\pm1.43}$ | $34.43^{\pm0.09}$ | $85.52^{\pm0.68}$ | $49.14^{\pm0.43}$ | $83.32^{\pm2.94}$ | $88.44^{\pm0.09}$ | $59.48^{\pm1.63}$ | $91.55^{\pm0.17}$ | $47.72^{\pm0.56}$ | 77.97 | 51.4 |
| | WPCL | $91.72^{\pm0.27}$ | $26.68^{\pm1.99}$ | $51.21^{\pm0.14}$ | $70.47^{\pm0.19}$ | $28.44^{\pm0.37}$ | $98.6^{\pm0.3}$ | $89.42^{\pm0.2}$ | $37.81^{\pm1.17}$ | $89.52^{\pm0.07}$ | $34.7^{\pm2.24}$ | 80.47 | 42.42 |
| | MSPCL | $93.82^{\pm0.81}$ | $27.41^{\pm2.42}$ | $51.73^{\pm0.31}$ | $71.79^{\pm0.42}$ | $8.67^{\pm0.37}$ | $99.68^{\pm0.15}$ | $97.33^{\pm0.02}$ | $11.48^{\pm0.46}$ | $92.86^{\pm0.39}$ | $31.47^{\pm2.24}$ | **83.94** | **35.54** |
| | CE | $91.73^{\pm0.5}$ | $30.98^{\pm5.21}$ | $32.15^{\pm0.15}$ | $83.9^{\pm1.49}$ | $10.6^{\pm0.29}$ | $98.77^{\pm0.14}$ | $35.64^{\pm0.86}$ | $90.04^{\pm0.89}$ | $62.31^{\pm0.23}$ | $59.58^{\pm0.81}$ | 46.49 | 72.65 |
| VGG19 | PCL | $87.45^{\pm0.94}$ | $62.83^{\pm4.07}$ | $37.56^{\pm0.33}$ | $70.55^{\pm0.38}$ | $76.03^{\pm0.21}$ | $39.3^{\pm0.96}$ | $46.24^{\pm0.52}$ | $98.41^{\pm0.59}$ | $71.24^{\pm0.28}$ | $88.81^{\pm0.7}$ | 63.7 | 71.98 |
| | WPCL | $93.59^{\pm0.26}$ | $29.45^{\pm3.24}$ | $35.26^{\pm0.04}$ | $77.44^{\pm0.8}$ | $57.24^{\pm0.58}$ | $82.11^{\pm2.61}$ | $91.59^{\pm0.07}$ | $26.41^{\pm0.28}$ | $91.21^{\pm0.09}$ | $27.95^{\pm0.94}$ | **73.78** | **48.67** |
| | MSPCL | $74.84^{\pm0.81}$ | $79.46^{\pm1.96}$ | $32.43^{\pm0.02}$ | $94.65^{\pm0.5}$ | $73.33^{\pm0.25}$ | $59.4^{\pm1.54}$ | $94.46^{\pm0.1}$ | $26.67^{\pm2.78}$ | $87.59^{\pm0.25}$ | $55.27^{\pm1.89}$ | 72.53 | 63.09 |
| CLIP | - | $53.87^{\pm3.81}$ | $86.44^{\pm0.40}$ | $43.60^{\pm6.39}$ | $95.99^{\pm4.01}$ | $56.10^{\pm12.17}$ | $91.46^{\pm5.29}$ | $61.34^{\pm0.0}$ | $93.91^{\pm0.0}$ | $50.00^{\pm0.0}$ | $88.38^{\pm4.41}$ | 52.98 | 91.24 |

Table S.1: The OOD detection results on the GTSRB dataset for the real-world OOD cases, the six trained DNNs, four training methods, and the six OOD detection methods. The OOD data used for this evaluation is a dataset consisting of 260 OOD images collected from the internet. The optimization methods include 1) the common cross-entropy (CE) loss; 2) the original PCL; 3) our WPCL; 4) and our MSPCL. The OOD detection methods include IF, class-based IF, KNN, Mahalanobis distance, ReAct, and ViM. The reported metrics are AUROC and false positive rate at 95 percept true positive rate. The last two columns on the right indicate the average over all the OOD detection methods on each row and each metric. Please note that the OOD detection methods results are averaged over 5 different random seeds used for optimizing the OOD detection methods, and the elevated numbers besides them are the standard deviation over the 5 run.

### S.2   Extra Details On The GTSRB Dataset

The results of detecting the real-world OOD samples on different training approaches on the GTSRB dataset are presented in Table S.1. This includes the AUROC and FPR for six individual OOD detection methods

| Model | | IF AUROC↑ | IF FPR↓ | KNN AUROC↑ | KNN FPR↓ | Mahal. AUROC↑ | Mahal. FPR↓ | ReAct AUROC↑ | ReAct FPR↓ | ViM AUROC↑ | ViM FPR↓ | Average AUROC↑ | Average FPR↓ |
|---|---|---|---|---|---|---|---|---|---|---|---|---|---|
| ResNet18 | CE | $80.51^{\pm0.0}$ | $79.32^{\pm0.0}$ | $88.97^{\pm0.0}$ | $50.54^{\pm0.0}$ | $23.07^{\pm0.0}$ | $98.6^{\pm0.0}$ | $89.91^{\pm0.23}$ | $39.2^{\pm0.97}$ | $89.05^{\pm0.08}$ | $51.14^{\pm1.99}$ | **74.3** | 63.76 |
| | PCL | $97.61^{\pm0.0}$ | $7.52^{\pm0.0}$ | $63.41^{\pm0.0}$ | $64.77^{\pm0.0}$ | $5.87^{\pm0.0}$ | $99.82^{\pm0.0}$ | $97.27^{\pm0.12}$ | $8.64^{\pm0.55}$ | $21.77^{\pm0.09}$ | $97.23^{\pm0.08}$ | 57.19 | **55.6** |
| | WPCL | $87.31^{\pm0.0}$ | $100.0^{\pm0.0}$ | $91.69^{\pm0.0}$ | $30.59^{\pm0.0}$ | $14.13^{\pm0.0}$ | $99.73^{\pm0.0}$ | $77.93^{\pm0.65}$ | $62.35^{\pm1.64}$ | $90.97^{\pm0.22}$ | $41.1^{\pm1.79}$ | 72.41 | 66.75 |
| | MSPCL | $89.42^{\pm0.0}$ | $36.58^{\pm0.0}$ | $77.48^{\pm0.0}$ | $67.16^{\pm0.0}$ | $7.88^{\pm0.0}$ | $99.82^{\pm0.0}$ | $76.04^{\pm0.36}$ | $71.91^{\pm0.79}$ | $90.05^{\pm0.19}$ | $43.65^{\pm0.35}$ | 68.17 | 63.82 |
| ResNet50 | CE | $76.44^{\pm0.0}$ | $75.18^{\pm0.0}$ | $85.15^{\pm0.0}$ | $56.8^{\pm0.0}$ | $28.95^{\pm0.0}$ | $95.5^{\pm0.0}$ | $81.02^{\pm0.04}$ | $57.38^{\pm0.51}$ | $90.35^{\pm0.08}$ | $36.88^{\pm0.54}$ | **71.77** | **65.24** |
| | PCL | $85.44^{\pm0.0}$ | $70.45^{\pm0.0}$ | $58.51^{\pm0.0}$ | $65.86^{\pm0.0}$ | $16.75^{\pm0.0}$ | $97.75^{\pm0.0}$ | $85.78^{\pm0.28}$ | $59.39^{\pm0.58}$ | $83.17^{\pm0.02}$ | $64.47^{\pm0.75}$ | 65.29 | 73.95 |
| | WPCL | $72.79^{\pm0.0}$ | $93.24^{\pm0.0}$ | $62.02^{\pm0.0}$ | $92.34^{\pm0.0}$ | $0.0^{\pm0.0}$ | $99.28^{\pm0.0}$ | $74.45^{\pm0.39}$ | $55.62^{\pm0.72}$ | $79.34^{\pm0.23}$ | $69.42^{\pm1.38}$ | 57.63 | 81.82 |
| | MSPCL | $87.57^{\pm0.0}$ | $45.99^{\pm0.0}$ | $74.69^{\pm0.0}$ | $80.77^{\pm0.0}$ | $1.02^{\pm0.0}$ | $99.64^{\pm0.0}$ | $50.0^{\pm0.0}$ | $60.9^{\pm1.21}$ | $53.91^{\pm0.07}$ | $56.52^{\pm0.71}$ | 52.62 | 69.65 |
| WideResNet50 | CE | $78.27^{\pm0.0}$ | $77.12^{\pm0.0}$ | $84.5^{\pm0.0}$ | $63.33^{\pm0.0}$ | $18.61^{\pm0.0}$ | $96.62^{\pm0.0}$ | $84.35^{\pm0.06}$ | $58.46^{\pm0.58}$ | $91.35^{\pm0.07}$ | $36.37^{\pm0.64}$ | **71.42** | **66.38** |
| | PCL | $67.36^{\pm0.0}$ | $66.26^{\pm0.0}$ | $39.24^{\pm0.0}$ | $66.76^{\pm0.0}$ | $23.89^{\pm0.06}$ | $93.11^{\pm0.17}$ | $66.07^{\pm0.28}$ | $58.99^{\pm0.86}$ | $84.79^{\pm0.08}$ | $55.82^{\pm0.42}$ | 54.84 | 68.3 |
| | WPCL | $82.34^{\pm0.0}$ | $92.93^{\pm0.0}$ | $72.48^{\pm0.0}$ | $73.56^{\pm0.0}$ | $44.1^{\pm0.01}$ | $96.12^{\pm0.04}$ | $50.0^{\pm0.0}$ | $70.59^{\pm1.25}$ | $51.76^{\pm0.24}$ | $78.5^{\pm0.3}$ | 60.14 | 82.34 |
| | MSPCL | $84.5^{\pm0.0}$ | $56.8^{\pm0.0}$ | $72.07^{\pm0.0}$ | $73.92^{\pm0.0}$ | $44.7^{\pm0.0}$ | $81.97^{\pm0.03}$ | $50.0^{\pm0.0}$ | $66.18^{\pm0.61}$ | $60.96^{\pm0.21}$ | $69.71^{\pm0.39}$ | 62.45 | 69.72 |
| ResNeXt50 | CE | $85.87^{\pm0.0}$ | $57.25^{\pm0.0}$ | $85.16^{\pm0.0}$ | $58.83^{\pm0.0}$ | $0.0^{\pm0.0}$ | $100.0^{\pm0.0}$ | $80.79^{\pm0.05}$ | $56.63^{\pm1.21}$ | $26.74^{\pm0.13}$ | $99.12^{\pm0.11}$ | 55.71 | 74.37 |
| | PCL | $94.57^{\pm0.0}$ | $41.04^{\pm0.0}$ | $64.96^{\pm0.0}$ | $65.45^{\pm0.0}$ | $14.35^{\pm0.0}$ | $99.28^{\pm0.0}$ | $94.16^{\pm0.12}$ | $29.56^{\pm0.72}$ | $91.08^{\pm0.08}$ | $78.68^{\pm0.62}$ | **71.82** | 62.8 |
| | WPCL | $59.04^{\pm0.0}$ | $86.31^{\pm0.0}$ | $54.33^{\pm0.0}$ | $93.42^{\pm0.0}$ | $50.88^{\pm0.0}$ | $97.43^{\pm0.0}$ | $50.21^{\pm0.04}$ | $79.77^{\pm0.45}$ | $43.04^{\pm0.16}$ | $94.13^{\pm0.11}$ | 51.5 | 90.21 |
| | MSPCL | $95.24^{\pm0.0}$ | $13.47^{\pm0.0}$ | $85.48^{\pm0.0}$ | $59.77^{\pm0.0}$ | $0.62^{\pm0.0}$ | $100.0^{\pm0.0}$ | $50.0^{\pm0.0}$ | $55.9^{\pm0.64}$ | $50.0^{\pm0.01}$ | $58.09^{\pm0.38}$ | 56.27 | **57.45** |
| VGG16 | CE | $94.19^{\pm0.14}$ | $21.2^{\pm0.9}$ | $34.95^{\pm0.04}$ | $72.89^{\pm0.71}$ | $15.12^{\pm0.09}$ | $98.35^{\pm0.08}$ | $93.31^{\pm0.14}$ | $21.29^{\pm0.43}$ | $92.3^{\pm0.14}$ | $23.76^{\pm0.85}$ | 78.69 | 34.78 |
| | PCL | $81.58^{\pm0.71}$ | $72.19^{\pm1.95}$ | $39.84^{\pm0.12}$ | $86.67^{\pm0.22}$ | $36.9^{\pm0.29}$ | $98.05^{\pm0.32}$ | $65.34^{\pm0.16}$ | $90.65^{\pm0.44}$ | $71.45^{\pm0.08}$ | $81.15^{\pm0.55}$ | 64.55 | 82.67 |
| | WPCL | $76.07^{\pm0.79}$ | $75.75^{\pm0.61}$ | $57.9^{\pm0.1}$ | $82.35^{\pm0.35}$ | $25.77^{\pm0.1}$ | $98.47^{\pm0.2}$ | $71.0^{\pm0.12}$ | $83.72^{\pm0.24}$ | $72.51^{\pm0.14}$ | $80.43^{\pm0.25}$ | 69.37 | 80.56 |
| | MSPCL | $95.34^{\pm0.49}$ | $24.74^{\pm3.65}$ | $57.54^{\pm0.2}$ | $70.07^{\pm0.13}$ | $6.29^{\pm0.07}$ | $99.93^{\pm0.07}$ | $97.68^{\pm0.03}$ | $9.03^{\pm0.15}$ | $92.79^{\pm0.21}$ | $27.96^{\pm1.44}$ | **85.84** | **32.95** |
| VGG19 | CE | $91.77^{\pm0.2}$ | $26.2^{\pm1.77}$ | $36.13^{\pm0.05}$ | $69.57^{\pm0.12}$ | $10.39^{\pm0.06}$ | $98.94^{\pm0.1}$ | $48.48^{\pm0.2}$ | $83.65^{\pm0.7}$ | $71.52^{\pm0.24}$ | $44.08^{\pm0.32}$ | 51.66 | **64.49** |
| | PCL | $83.22^{\pm0.55}$ | $60.75^{\pm0.79}$ | $40.12^{\pm0.12}$ | $78.47^{\pm0.34}$ | $50.6^{\pm0.48}$ | $98.64^{\pm0.33}$ | $43.62^{\pm0.15}$ | $99.24^{\pm0.24}$ | $64.99^{\pm0.28}$ | $78.3^{\pm0.95}$ | 56.51 | 83.08 |
| | WPCL | $89.77^{\pm0.36}$ | $57.29^{\pm1.99}$ | $41.32^{\pm0.05}$ | $71.18^{\pm0.12}$ | $41.28^{\pm0.21}$ | $97.25^{\pm0.23}$ | $83.79^{\pm0.06}$ | $76.66^{\pm0.27}$ | $83.4^{\pm0.1}$ | $77.21^{\pm1.2}$ | **67.91** | 75.92 |
| | MSPCL | $79.04^{\pm1.03}$ | $70.16^{\pm1.85}$ | $40.53^{\pm0.12}$ | $79.59^{\pm0.72}$ | $45.09^{\pm0.25}$ | $87.2^{\pm0.76}$ | $76.7^{\pm0.04}$ | $91.71^{\pm0.78}$ | $77.67^{\pm0.17}$ | $79.33^{\pm0.65}$ | 63.81 | 81.6 |

Table S.2: The OOD detection results on the GTSRB dataset for the RP2 adversarial OOD cases, the six trained DNNs, four training methods, and the six OOD detection methods. The OOD data used for this evaluation of the attacked images using the RP2 real-world adversarial sticker attack Eykholt et al. (2018). The optimization methods include 1) the common cross-entropy (CE) loss; 2) the original PCL; 3) our WPCL; 4) and our MSPCL. The OOD detection methods include IF, class-based IF, KNN, Mahalanobis distance, ReAct, and ViM. The reported metrics are AUROC and false positive rate at 95 percept true positive rate. The last two columns on the right indicate the average over all the OOD detection methods on each row and each metric. Please note that the OOD detection methods results are averaged over 5 different random seeds used for optimizing the OOD detection methods, and the elevated numbers beside them are the standard deviation over the 5 run.

and their averages on each training type of each DNN, wherein we have implemented and evaluated a class-wise isolation forest method to better understand the clustering capability of the standard isolation forest, which is not presented in the main paper. Similarly, similar results are presented for the OOD data generated by the RP2 adversarial attack in Table S.2. For both of these tables, the OOD detectors are fitted 5 times. Therefore, the results are the average results achieved over the 5 iterations, along with their standard deviation, as shown on the side of each number.

## S.3 Comparison With Contrastive Language–Image Pre-training (CLIP)

We have also conducted a comparison between our trained DNNs and the Contrastive Language–Image Pre-training (CLIP) Radford et al. (2021), which is a language vision model trained on web-scale data. However, as it is not specially trained in traffic sign classification, we limited the OOD detection to our real-world OOD data subset. For this comparison, we utilized a CLIP with a ResNet50 backbone. The results are presented in Table S.1.

## S.4 Videos

Although visualizing latent space activations of DNNs accurately is not easily feasible due to the high number of dimensions of such data, we have visualized a 3 dimensional version of the activations to analyze the effect of different training methods on the image encoding done by the CNN backbones of our DNNs. Accordingly, we have extracted the activations generated by the WideResNet50 DNN on the whole GTSRB test dataset, which consists of more than 12000 images. These activations are taken from the latest average pool layer from the WideResNet50 DNN. Following that, we have reduced the number of dimensions of these data to 3 using the principal component analysis (PCA). Furthermore, we have visualized this using the Plotly Inc. (2015) visualization tool, which leads to web-based interactive plots. Finally, we have captured these plots in our supplementary video, which consists of four parts for the four training methods we conducted, including

the cross-entropy loss, PCL, our WPCL, and our MSPCL. In this video, the colors indicate classes, wherein due to a high number of classes, some are visualized with similar colors while still being distinguishable due to their distances.

Moreover, we would like to emphasize the fact that our videos are generated solely for visualization purposes and do not serve as the basis for qualitative analysis. However, one can observe that different training methods do affect the way the image data samples are encoded by the CNN backbone.

