# OpenReview forum: "Towards Prototype Conformity Loss Functions for Better Outlier Detection in Traffic Sign Image Classification"
_TMLR — Rejected by TMLR_

### Review · Reviewer_7Y8M · 2024-03-15

**Summary Of Contributions:**

This paper focuses on OOD detection in traffic sign image classification. Specifically, this paper argues that the original cross-entropy loss can not semantically disentangle similar classes in convolutional neural networks. Thus, this work adopts the prototype conformity loss function for better OOD detection. Furthermore, this work proposes two variations, e.g., weighted and multi-scale PCL to enhance the disentangling of different classes. Various experiments are conducted to demonstrate the effectiveness of the algorithm design.

**Audience:**

Yes

**Claims And Evidence:**

No

**Requested Changes:**

Please refer to the weaknesses part for specific comments on revision.

**Strengths And Weaknesses:**

Strengths:
1. This paper studies a practical research problem and shows the method's benefits of applying it to real-world traffic sign image classification tasks.
2. This work applies the contrastive PCL loss function from literature to enhance the feature-based OOD detection algorithms. In addition, it also provides two variants, e.g., weighted prototype conformity loss WPCL and multi-scale prototype conformity loss MSPCL to tackle the problem of very similar classes disentanglement.
3. Various experiments are conducted to demonstrate the effectiveness of the proposed methods.

Weaknesses:
1. The background and rigorous definition of OOD data is missing in the current version, seeming that the "OOD data" also includes the adversarial data in consideration. It is sometimes confusing if there is no such clear definition of the current research task. Please refer to [1] to give a clear introduction for the current research problem setting, and provide the detection target for what kind of OOD data is this work focused on.
2. Specifically, adversarial data and normal OOD data have different definitions and are tackled with different methodologies in the literature [1]. Without a specific target statement as the previous weakness part mentioned, it is unclear why use some methods of normal OOD detection to detect those samples with covariant shift. So please clearly explain the detection setups and the motivation, and revise some terms accordingly.
3. For the proposed method, the specific motivation for weighted PCL and multi-scale PCL is not empirically verified. Specifically, there is no empirical evidence to show that the visual similarity of certain classes would minimize the effect of the contrastive proximity, and there is no clear definition of the visual similarity. It would also be better to visualize the overlapping hyper-balls of similar classes.
4. The current version didn't clearly explain why such a design of WPCL or MSPCL can benefit the detection tasks.
5. The experimental results show sometimes the WPCL or MSPCL can even degrade the detection performance across different model structures and detection methods, it needs further discussion on the failure cases to provide a thorough understanding of the proposed methods.

[1] Generalized Out-of-Distribution Detection: A Survey.

---

> ### Author Response · Authors · 2024-04-28
> **Our answers to your review**
>
> While we appreciate your efforts in providing such a detailed and helpful review, we have done our best to adopt our paper based on your suggestions whenever it was possible and have provided the following answer to each of your points in the following:
>
> 1- In automated driving applications, where infinite combinations of scenarios and events can face the AD vehicles, it is important to rely on the system's known and seen scenarios to prevent malfunction in unknown scenarios. Having this motivation in mind, we considered all the applied augmentations and attacks as well as the real-world unseen data samples as OOD. In other words, any data sample that is inferred to the DNN and is not within its operational design domain (ODD) can be considered OOD in this setup.
>
> 2- We appreciate the suggestion and find it insightful. Therefore, we added a new paragraph in the introduction section to explain our motivation for such a design, defined the scope of OOD data we would aim to tackle, and updated the title of the paper according to 'of distribution instead' of 'outlier,' all based on the definition provided by your suggested paper.
>
> 3- We have performed a dispersion analysis on the WideResNet50 DNN on the GTSRB dataset, and the results are now presented in the newly added Figure 9. There, it is evident that in such a particular case, the inter-class distance is higher for WPCL while the intra-class variation and std are lower, indicating a latent representation with compact clusters projected far from each other for the predefined groups of similar classes.
>
> 4- We provided various theoretical and experimental illustrations along with explanations in the introduction and the method section to emphasize the need and benefit of our proposed variations of PCL. Please refer to Figures 1, 3, and the newly added Figure 9, wherein we demonstrate the effect of each optimization technique on the projection of those data samples into the latent space and their proximity. Especially in Figure 9, it is evident that the original WPCL and MSPCL reduce the negative effect of the original PCL in projecting similarly shaped classes into different dense classes with higher inter-class distance.
>
> 5- The noticeable detection performance degradations happened for two of the DNNs (ResNext50 and VGG16) on CURE-TSR, which are discussed in the revision. Our comparison of the two datasets with the same domain aimed to provide evidence that our methods help in scenarios where multiple visually similar classes are present, which is only the case for the GTSRB dataset, as the CURE-TSR dataset has groups of similar classes as single classes.

---

### Review · Reviewer_9vh7 · 2024-03-17

**Summary Of Contributions:**

The work proposed the adoption of prototype conformity loss (PCL) in the training of classifier in safety-critical tasks related to self-driving cars. PCL regularize the classifier by encouraging outputs from DNNs of the same class to be close while pulling the outputs from different classes away from each other. Two variants of PCL are introduced by the work: weighted PCL and multi-scale PCL. Weighted PCL applies additional weighting to the terms that pull representations of different classes away from each other in the original PCL. Multi-scale PCL applies grouping to the class labels of data and defines PCL at both the class and group levels. Experiment results shows model training with WPCL or multi-scale PCL improves the out-of-distribution (OOD) under various settings.

**Audience:**

Yes

**Broader Impact Concerns:**

The work doesn't discussed its limitations or potential broader social impacts. As a work focusing on safety-critical application, it should include discussions on its limitations and potential broader impacts.

**Claims And Evidence:**

Yes

**Requested Changes:**

1. Can the author also shows OOD detection results on models that trained with both WPCL and multi-scale PCL? It seems that they could be complimentary to each other and might deliver the best results.
2. Contrastive learning seems to have similar effect as the PCL by encouraging the representations of different views of a sample to stay close to each other while pulling the representations of different data away from each other. Existing works [1] also shows the contrastive learning could improve OOD detection performance. The author should consider comparing the proposed WPCL and multi-scale PCL against contrastive learning.
3. Many model pre-trained model on web-scale vision-language data like CLIP[2]  also show the capability of zero-shot OOD detection[3]. I would encourage the author to also report the OOD performance of such models.

[1] Winkens, Jim, et al. "Contrastive training for improved out-of-distribution detection." arXiv preprint arXiv:2007.05566 (2020)

[2] Radford, Alec, et al. "Learning transferable visual models from natural language supervision." International conference on machine learning. PMLR, 2021.

[3] Esmaeilpour, Sepideh, et al. "Zero-shot out-of-distribution detection based on the pre-trained model clip." Proceedings of the AAAI conference on artificial intelligence. Vol. 36. No. 6. 2022.

**Strengths And Weaknesses:**

Strengths:
1. The proposed approach is generally applicable to different models on a topic of important practical value.
2. The adaptation of the PCL are well motivated.
2=3. The evaluation of the work is comprehensive by considering combing different models and OOT detection method with the proposed regularization term. Experiment results shows that WPCL or multi-scale PCL delivers the best results in most cases.

Weakness:
1. The novelty of the work with the proposed two variants of PCL is marginal.
2. Please see requested changes.

---

> ### Author Response · Authors · 2024-04-28
> **Our answers to your comments**
>
> We would like to also appreciate your feedback and suggestions. We have implemented your suggestions as much as possible and provided better explanations and analysis, hoping to enrich the readability and comprehensiveness of our paper. Here are our answers to your comments and questions:
>
> 1- We appreciated the proposal and implemented it accordingly. Due to time constraints, we were able to finish the experiments on the GTRSB dataset.  Table 2 is updated with the new results. We aim to provide the results for the CURE-TSR dataset in the meantime as well.
>
> 2- Due to our time constraints and the extensive experiment setup of our research project, we could not implement your suggestion. However, as you and the other reviewers also noticed, PCL is similar to other contrastive learning approaches. Therefore, we would encourage applying our grouping paradigm in such other approaches to study their effects. We discussed this limitation in the revised version.
>
> 3- We have implemented the clip into our OOD pipeline using the pre-trained weights provided by the authors and tested the OOD performance in the real world and the data samples we collected. The results are presented in the supplementary section in Table S.1, which indicates that our approaches achieved better OOD detection performance.
>
> Broader Impact Concerns: as mentioned above, we provided a more detailed discussion on the limitations and suggestions for future work in the revised version.

---

### Review · Reviewer_fqEa · 2024-04-17

**Summary Of Contributions:**

The paper introduces an optimization-based approach aimed at enhancing OOD detection in traffic sign image classification. The authors claim that incorporating an additional PCL loss leads to superior results in OOD detection, while maintaining classification performance without noticeable drop. However, from my perspective, the experimental data and metrics presented in the paper may lack sufficient evidence to support their claims.

**Audience:**

Yes

**Broader Impact Concerns:**

None.

**Claims And Evidence:**

No

**Requested Changes:**

Reasonable explanations on Figure 5 & 6 and Table 2 & 3 & S.1 & S.2 are necessary to support the claimed contributions in the paper. I recommend the authors modify their paper according to the items listed in weakness section.

**Strengths And Weaknesses:**

### Strengths

* The proposed method is simple and easy to understand and implement.
* The paper presents rich experimental results listed in tables.

### Weaknesses
* The paper focuses on the effects of an additional loss and its variations on OOD detection.  However, such contrastive losses are already widely used in various machine learning tasks, diminishing the novelty of the proposed method.
* One potential limitation of the proposed method is that it is difficult to apply it in any pretrained model. Additionally, the paper set a strong limitation on its orientated task, namely traffic sign image classification.
* Figure 5 & 6 fail to clearly demonstrate the advantage of proposed method by using “more separated distributions and thereby improve OOD detection”.
* Furthermore, Table 1 does not provide sufficient support for the claim that the classification performance remains unaffected by the proposed method. A drop of over 5% in metrics, such as precision/recall, should not be disregarded. For instance, see the P/R result of WPCL/MSPCL on CURE-TSR.
* Although the paper presents extensive experimental data in several tables, they are not pervasive enough to support the overall claims. For example, in Table 2 & 3 & S.1 & S.2, the average performance on different networks or different OOD detection method is used as main evidence. However, averages can be misleading, as extreme values can skew the conclusions. In my opinion, the proportion of the method that is effective in different experimental settings is more valuable. The proposed method does not appear to be advantageous using such metric. Moreover, the significant performance variation of proposed 3 methods (PCL/ WPCL/MSPCL) adds difficulty in selecting the most suitable approach.

---

> ### Author Response · Authors · 2024-04-28
> **Our answers to your comments**
>
> We would also like to appreciate your comments and suggestions. We aimed to enrich our paper further based on your feedback, which is included in our submitted revisions. Here, we would also like to provide an answer to all of the points raised during your review:
>
> 1- Our research aims primarily to close the gap between the SOTA research and their applicability as well as their adaptability to real-world settings. In safety-critical applications such as automated driving, where active human monitoring after deployment is impossible, having monitoring systems that can raise warnings of potential hazards is crucial in order to prevent hazardous events. In this paper, we proposed methods to adopt an already proven contrastive approach in a real-world setting to demonstrate its effectiveness.
>
> 2- We report in our experiments that various SOTA OOD detection methods can be boosted by our methods, i.e., IF, React, and VIM were boosted by 9.95%, 8.03%, and 5.4% on average, respectively, and up to 40% in extreme cases, i.e. Combined approach (new suggested by the reviewers) on ResNeXt50 with VIM.  In fact, this, alongside the minor difference in performance changes, demonstrates the applicability of our methods in helping the SOTA OOD detectors coexist with the underlying DNNs and act as their monitoring modules. While we had not studied the effect of our approach when only used as fine-tuning with limited iterations, we suggest that this could also be potentially applicable.
>
> 3- We have updated both of the figures with RP2 and real OOD results and provided a more minimal design alongside the threshold line to showcase the separability power of MSPCL on ResNet50, which helps IF to predict with superior performance. These figures are in coordination with the Tables S.1 and S.2.
>
> 4- The average accuracy result of the MSPCL on the CURE-TSR dataset was mistakenly typed as 74.74%, which was updated with the correct number as 94.77. Furthermore, we would argue that in order to proof the generality of our method to different architectures with different depths, we decided to extend our experiment to twelve training setups -six DNNs and two datasets-, which led to not tuning the hyperparameters per each training setup effectively. While one could search for better training parameters to achieve the best performance based on each model, we relied on an aggregated conclusion based on all the training setups together, wherein except for VGG16 and ResNeXt50 on the CURE-TSR dataset, all the other training setups led to minor drops in performance and in many other setups minor to a major boost in performance across different metrics with either of the PCL variants, e.g. WideResNet50 on CURE-TSR with +11 in Precision and +6 in Recall.
>
> 5- We provided two new Figures to provide a more in-depth analysis of the methods and OOD data combinations. Figures 5 and 6 in the revision submitted. We have also provided discussions according to these new digures.

---

### Decision · Action_Editor_jSAG · 2024-05-26

**Recommendation:** Reject

**Comment:**

This paper tackles the problem of out-of-domain detection for traffic sign detection. This paper argues that the original cross-entropy loss can not semantically disentangle similar classes for neural networks. The authors proposed adopts the prototype conformity loss function for better OOD detection. Two variants of the loss functions were proposed, including the weighted and multi-scale PCL to enhance the disentangling of different classes. Experiments are conducted to demonstrate the effectiveness of the algorithm design.

The AE doesn't think the paper is ready for publication in TMLR yet. The paper has the following limitations and prevents it from being accepted:
- The scope of OOD in just traffic sign classification is really narrow and will attract quite limited (if not none) TMLR audience. There are already quite some OOD benchmarks on general image classification out there. The authors are recommended to consider a larger scope of tasks to test the proposed losses but not limited to only traffic sign classification;
- The work doesn't provide a clear and rigorous definition of OOD data in the context of traffic sign classification;
- The authors claimed theories of the proposed losses but failed to adequately deliver them;
- The rebuttal is short in general. Quite some reasonable requests of the reviewers were not fully fulfilled during the rebuttal.

**Audience:**

Yes.

**Claims And Evidence:**

Not entirely. The OOD data in the context of traffic sign classification are not well defined.